# Elucidating Guidance in Variance Exploding Diffusion Models: Fast Convergence and Better Diversity

## Abstract

Recently, the conditional diffusion models have shown an impressive performance in many areas, such as text-to-image, 3D, and video. To achieve a better alignment with the given condition, guidance-based methods are proposed and become a standard component of diffusion models. Though the guidance-based methods are widely used, the theoretical guarantee mainly focuses on the variance-preserving (VP)-based models and lacks current state-of-the-art (SOTA) variance-exploding (VE) models for conditional generation. In this work, for the first time, we elucidate the influence of guidance for VE models and explain why VE-based models perform better than VP models in the context of Gaussian mixture models from classification confidence and diversity perspectives. For the classification confidence, we prove the convergence rate for the confidence w.r.t. the strength of guidance $\eta$ for VE models is $1 - \eta^{-1}(\log \eta)^2$, which is faster than $1 - \eta^{-e^{-T}}(\log \eta)^{2e^{-T}}$ result for VP models ($T$ is the diffusion time). This result indicates that the VE models have a stronger ability to align with the given condition, which is important for the conditional generation. For the diversity, previous works show that when facing strong guidance, VP models tend to generate extreme samples and suffer from the mode collapse phenomenon. However, for VE models, we show that since their forward process maintains the multi-modal property of data, they have a better ability to avoid the mode collapse facing strong guidance. The simulation and real-world experiments also support our theoretical results.

## 1 Introduction

Recently, diffusion models have shown an impressive performance in generating diverse, high-quality samples and show state-of-the-art performance in many areas, such as 2D, 3D, video generation (Rombach et al., 2022; Ho et al., 2022; Chen et al., 2023; Ma et al., 2024; Chen et al., 2024; Long et al., 2024). In these areas, the users give a class label or text prompt (the condition $y$), diffusion models aims to generate samples with the given condition $y$. To better align with the given condition, the guidance-based methods, including classifier guidance (Dhariwal and Nichol, 2021) and classifier-free guidance (CFG) method Ho and Salimans (2022), are proposed and have been integrated into diffusion models as a standard operation.

There are two common diffusion models: the variance preserving (VP)-based models (Song et al., 2020) and the variance exploding (VE)-based models (Song et al., 2020; Karras et al., 2022). The diffusion process of VP-based models corresponds to an Ornstein-Uhlenbeck process, and the stationary distribution is $\mathcal{N}(0, \mathbf{I})$. The diffusion process of VE-based models has an exploding variance. Two famous VE-based models are VE (SMLD) (Song et al., 2020) and VE (EDM) (Karras et al., 2022), where the first achieves a competitive performance with VP-based models, and the latter achieves SOTA performance (Karras et al., 2024a;b) and is widely used in the one-step generation task. The basic and important guidance methods, such as the classifier guidance method and the CFG method, are proposed based on the VP-based models and provide an important boost in the conditional sampling task. Then, the VE-based diffusion models VE (EDM) with guidance have recently achieved SOTA performance in conditional image generation (Karras et al., 2024a;b).

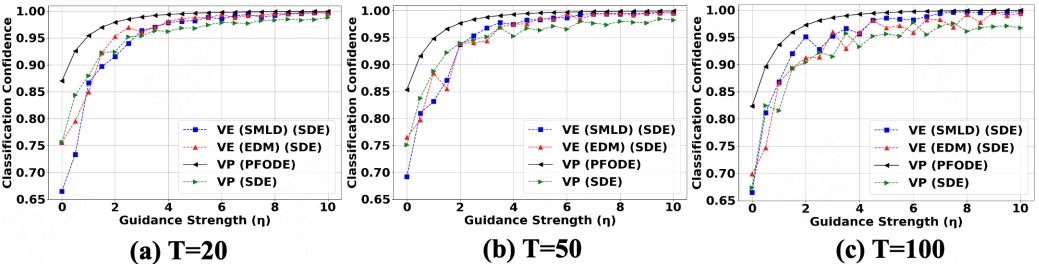

Figure 1: The Influence of Guidance for Classification confidence (VP, VE (SMLD) and VE (EDM)). The convergence rate for VP (SDE) (green line) is much slower than other three lines.

Despite the great performance of guidance-based models in different diffusion models, the theoretical insight is lacking and focuses on the VP-based models (Wu et al., 2024; Bradley and Nakkiran, 2024; Chidambaram et al., 2024; Guo et al., 2024; Li and Jiao, 2025). Some works show the relationship between the guidance-based method and other methods, such as the predictor-corrector framework (Bradley and Nakkiran, 2024) and the first-order optimization (Guo et al., 2024). More recently, some works focus on the influence of the strength parameters of guidance $\eta$ from the classification confidence and diversity for VP-based models (Wu et al., 2024; Chidambaram et al., 2024; Li and Jiao, 2025). For the classification confidence, Wu et al. (2024) show that the convergence rate w.r.t. the strength of guidance $\eta$ is different for the stochastic (reverse SDE) and deterministic (reverse probability flow ODE, PFODE) sampling process, and the deterministic sampling process enjoys a faster convergence rate. For the diversity, Wu et al. (2024) and Chidambaram et al. (2024) show that strong guidance will lead to mode collapse and sample extreme samples in the support of the conditional distribution for VP-based models. Though these works make an important step in understanding the effect of guidance, the theoretical exploration of guidance for VE-based models is still lacking, and we can not explain why the VE-based models achieve great performance in the conditional generation. Therefore, the following natural question remains open:

*What is the role of guidance for VE-based models? Why VE (EDM) with guidance achieve SOTA performance in the conditional generation?*

### 1.1 OUR CONTRIBUTION

In this work, for the first time, we elucidate the role of guidance for VE-based models from the classification confidence and diversity perspective. From the classification confidence perspective, we prove the convergence rate w.r.t. the strength of guidance $\eta$ for VE-based models and show it is faster than the one for VP-based models. For the diversity, we intuitively explain why VE-based models can maintain the multi-modal property, meanwhile VP-based models collapse. Based on these results, we make the first step to explain the success of VE (EDM) with guidance in the conditional generation. The simulation and real-world experiments also support our theoretical results.

**Classification Confidence for VE: Poor Beginning, Fast Improvement.** As a start, we first study the classification confidence (Eq.3) of conditional diffusion models without guidance (Eq. 1) and prove that VP has the smallest error term $\exp(-T)$, where $T$ is the diffusion time. On the contrary, VE (EDM) has a polynomial error term $1/\sqrt{T}$ and VE (SMLD) suffers a larger $1/T$ error, which indicates that VE models have a worse performance compared with VP models without guidance.

When generating conditional samples, diffusion models usually add additional guidance with strength $\eta \geq 1$ to guarantee the alignment with the given condition. Hence, as the next step, we study the convergence guarantee w.r.t. to $\eta$ for the VE-based models under the stochastic and deterministic sampling processes. When considering the stochastic sampling process, the convergence guarantee is still $1 - \eta^{-1}(\log \eta)^2$ for the VE-based models. On the contrary, the VP-based models suffer a significantly slower $1 - \eta^{-e^{-T}}(\log \eta)^{2e^{-T}}$ convergence guarantee, which is heavily influenced by diffusion time $T$. This result indicates that the VE-based models have a stronger ability to align with the given condition, which leads to great performance in conditional generation. Our simulation experiments also exactly support the above discussion (Figure 1).

We also prove the $1 - \eta^{-1}(\log \eta)^2$ result for the VE-based models under the deterministic sampling process, which is the same as the results of VP models (Wu et al., 2024). We note that this results also

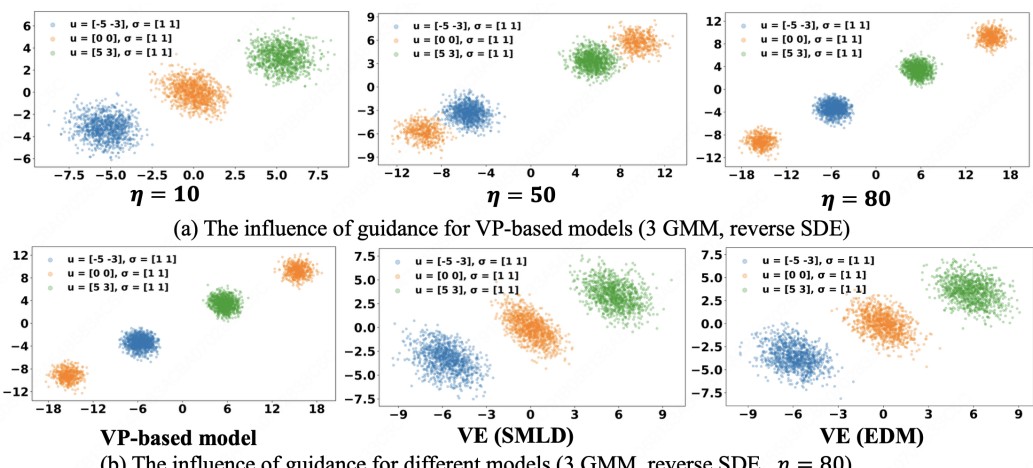

(a) The influence of guidance for VP-based models (3 GMM, reverse SDE)

(b) The influence of guidance for different models (3 GMM, reverse SDE , $\eta = 80$)

Figure 2: The Influence of Guidance for Multi-modal Property. VP-based models can not maintain the correct multi-modal property facing strong guidance $\eta$.

matches the empirical observation that the deterministic sampling process has a great performance in the conditional generation [1].

**VE Maintain Multi-modal Property Facing Strong Guidance.** For VP-based models, a higher classification confidence usually leads to lower diversity and tends to generate extreme samples in the support of the conditional distribution (Wu et al., 2024; Chidambaram et al., 2024). As shown in Figure 2 (a), with strong guidance, VP-based models with guidance can not generate the central modal (the orange one) and lose the diversity. On the contrary, as shown in Figure 2 (b), the VE-based models can alleviate the mode collapse phenomenon when facing strong guidance. We intuitively explain why VE-based models perform better in maintaining the multi-modal property by analyzing the property of their forward process. More specifically, the diffusion process of VP gradually removes the multi-modal information from data. On the contrary, the VE models maintain the multi-modal property during the diffusion process. Since the sampling process is obtained by reversing the diffusion process, this property holds for the sampling process, which leads to a better multi-modal ability for VE models facing strong guidance.

The above results show that VE (EDM) has a faster convergence rate than VP models and maintains the multi-modal property of the target distribution. Compared with VE (SMLD), the classification confidence for VE (EDM) is higher for $\forall \eta \geq 1$ and both maintain the multi-modal property. Hence, this work makes the first step to explain why VE (EDM) achieves great performance.

## 2 RELATED WORK

**Theory on Conditional Diffusion Models.** There is a series of works to study the conditional diffusion models (without guidance) from the estimation, reward improvement, and optimization perspective (Fu et al., 2024; Hu et al., 2024; Yuan et al., 2023; Guo et al., 2024). For the estimation error, Fu et al. (2024) and Hu et al. (2024) provide the minimax estimation results for conditional diffusion models with deep ReLU and diffusion transformer (DiT), respectively. Yuan et al. (2023) study the influence of high reward conditions under the linear subspace assumption and show the balance between the high reward and the off-support error. Guo et al. (2024) link the condition and the regularized optimization problem and provide the convergence guarantee for the iteration number.

**Theory on Guidance Diffusion Models.** There are only a few works analyze the role of additional guidance of conditional diffusion models and focus on the VP-based models. More specifically, Bradley and Nakkiran (2024) show the relationship between the classifier-free guidance and the predictor-corrector framework. Wu et al. (2024) consider a mixture of Gaussian distribution and prove the convergence guarantee for the classification confidence w.r.t. the strength of guidance $\eta$ under the reverse SDE and PFODE setting for VP-based models. Their results also show that

---

[1]For the sake of clarity, we do not present model experiments with deterministic samplers for VE (SMLD) and VE (EDM) in Figure 1. We provide these results in experiments in Appendix B, which fast converge to a high classification confidence and match our theoretical results.

the convergence guarantee under the reverse SDE setting is slower than the reverse PFODE setting for the VP-based models. Chidambaram et al. (2024) use a mixture of bounded data and show that with a very large $\eta$, diffusion models tend to generate extreme samples in the support of the conditional distribution. Li and Jiao (2025) analyze the role of guidance under a general data and prove that guidance preferentially enhances the generation of samples associated with higher classifier probability. Different from the above analysis focusing on VP-based models, this work aims to explain why VE-based models can achieve great performance in the conditional generation. For the VE-based models with additional guidance, Li et al. (2025) analyze the CFG method based on the linear diffusion models family (corresponds to the Gaussian distribution) and explain why naive conditional sampling is not enough by carefully analyzing each component of the CFG method. However, Li et al. (2025) do not analyze the convergence rate w.r.t. the guidance strength $\eta$ and their setting relies heavily on the linear diffusion models setting and Gaussian target data.

## 3 PRELIMINARIES

First, we introduce the basic knowledge of conditional diffusion models without guidance and discuss different diffusion models. Then, Section 3.1 introduces two widely used guidance methods: the classifier guidance and classifier-free guidance.

Let $p^*$ be the target distribution over $(x, y)$, where $x \in \mathbb{R}^d$ is the data (such as images) and $y$ is the corresponding data label. The conditional diffusion models aim to sample from the conditional distribution $p_*(x|y)$ when given a label $y$. Conditional diffusion models consist of two processes: the forward and reverse processes. The forward diffusion process gradually converts the conditional distribution to pure Gaussian noise, and the corresponding reverse process removes noise from pure Gaussian step by step to generate samples from the conditional distribution.

**General Forward Process.** The general forward process $\{p_t\}_{t \in [0, T]}$ has the following form:

$$\mathrm{d}z_t^{\rightarrow} = -f(t)z_t^{\rightarrow} \, \mathrm{d}t + g(t)\mathrm{d}B_t, \quad z_0^{\rightarrow} \sim p_*(\cdot|y) \in \mathbb{R}^d,$$

where $f(t)$ and $g(t)$ is non-negative non-decreasing sequence and $(B_t)_{t \geq 0}$ is a $d$-dimensional Brownian motion. After determining a forward process and given $z_0^{\rightarrow}$, the forward process conditional distribution $z_t^{\rightarrow}|z_0^{\rightarrow}$ is exactly $\mathcal{N}\left(m_t z_0^{\rightarrow}, \sigma_t^2 I_d\right)$, where $m_t$ and $\sigma_t^2$ is determined by $f(t)$ and $g(t)$.

There are two typical forward processes (Song et al., 2020): (1) Variance exploding (VE) SDE and (2) variance preserving (VP) SDE. When $f(t) = 1$ and $g(t) = \sqrt{2}$ [2], the process is instantiated as VPSDE, whose stationary distribution is $\mathcal{N}(0, I)$ with $m_t = e^{-t}$ and $\sigma_t^2 = 1 - e^{-2t}$. When the process only contains a diffusion term $g(t) = \sqrt{\mathrm{d}\sigma_t^2/\mathrm{d}t}$ and $f(t) \equiv 0$, the process is instantiated as VESDE with $m_t = 10, \forall t \in [0, T]$. Two common VE-based models are VE (SMLD) with $\sigma_t^2 = t$ (Song et al., 2020) and VE (EDM) with $\sigma_t^2 = t^2$ (Karras et al., 2022). In the early years, the conditional generation methods are proposed mainly based on VP models, and the VE (EDM) achieve SOTA performance in the conditional diffusion models. Since previous theoretical works for conditional diffusion models mainly analyze VP models, we focus on VE-based models and aim to explain why VE (EDM) performs well in conditional generation.

**Two typical Reverse Processes.** To generate samples from the conditional distribution, diffusion models reverse the forward process and obtain the reverse process:

$$\mathrm{d}z_t^{\leftarrow} = \left[f(T-t)z_t^{\leftarrow} + \frac{1+\alpha^2}{2}g(T-t)^2\nabla \log p_{T-t}\left(z_t^{\leftarrow}|y\right)\right]\mathrm{d}t + \alpha g(T-t)\,\mathrm{d}B_t, \quad (1)$$

where $z_0^{\leftarrow} \sim p_T(\cdot|y)$, $(z_t^{\leftarrow})_{t \in [0,T]} = \left(z_{T-t}^{\rightarrow}\right)_{t \in [0,T]}$ and $\alpha \in [0, 1]$. Since the reverse process has the same marginal distribution $p_t$ as the corresponding forward process, diffusion models can run the above process to generate the conditional distribution $p_*(\cdot|y)$ with the conditional score function $\nabla \log p_{T-t}\left(z_t^{\leftarrow}|y\right)$ (Song et al., 2020). For parameter $\alpha$, it is used to determine the type of the reverse process. When $\alpha = 0$, it is a deterministic process, called the probability flow ODE (PFODE). When $\alpha = 1$, this process is a stochastic process, which is called reverse SDE. Due to the additional randomness, the reverse SDE usually tends to generate more diverse samples. On the contrary, the reverse PFODE trends to generate samples align more closely with the given condition.

---

[2]We note that the VPSDE forward process allows $f(t) = \beta_t$ and $g(t) = \sqrt{\beta_t}$ with a bounded $\beta_t$. In this work, we adopt the choice $\beta_t = 1, \forall t \in [0, T]$ of Wu et al. (2024).

## 3.1 GUIDANCE-BASED DIFFUSION MODELS

Recent works add additional guidance to the score function to generate samples aligned with the target label (condition). There are two common guidance methods: classifier guidance and classifier-free guidance. In this work, for the sake of simplicity, we write $(z_t)_{0 \leq t \leq T} = (z_t^{\leftarrow})_{0 \leq t \leq T}$ and use $x_t$ instead of $z_t$ when adding additional guidance to the diffusion models.

**Classifier Guidance.** The classifier guidance method trains an additional classifier and adds the gradient of the logarithmic prediction probability of the classifier to the conditional score function to guide the diffusion models to generate data with given $y$ (Dhariwal and Nichol, 2021):

$$\mathrm{d}x_t = \left[ f(T-t)x_t + \frac{1+\alpha^2}{2} g(T-t)^2 (s_{T-t}(x_t, y) + \eta \nabla \log c_{T-t}(x_t, y)) \right] \mathrm{d}t \\ + \alpha g(T-t)\mathrm{d}B_t, \quad (2)$$

where the integer $\eta \geq 0$ is the strength of the guidance, $s_{T-t}(x, y)$ is an estimation of $\nabla \log p_{T-t}(x|y)$ and $c_{T-t}(x, y)$ is a probability classifier to estimate the conditional probability $p_{T-t}(y|x)$.

**Classifier-free Guidance.** Though the classifier guidance method provides an important boost in developing text-to-image generation, this method requires training an additional classifier and makes the training process more complex. To address this problem, the CFG method is proposed, which jointly trains a score $s_t(x, y)$ containing $x$ and $y$ and uses the following process to generate samples:

$$\mathrm{d}x_t = \left[ f(T-t)x_t + \frac{1+\alpha^2}{2} g(T-t)^2 ((1+\eta)s_{T-t}(x_t, y) - \eta s_{T-t}(x_t, \emptyset)) \right] \mathrm{d}t + \alpha g(T-t)\mathrm{d}B_t.$$

We note that when having access to the ground-truth functions $s_t(x, y) = \nabla_x \log p_t(x|y)$, $s_t(x) = \nabla_x \log p_t(x)$ and $c_t(x, y) = p_t(y|x)$, we can verify that $x_t$ of the above two methods is exactly the same when starting from the same initialization distribution (including $p_T(\cdot|y)$ and pure Gaussian $\mathcal{N}(0, \sigma_T^2 I)$). In this work, we adopt the Gaussian mixture models, whose ground-truth functions have a closed form, to analyze different diffusion models. With this target distribution, we aim to explain the different performance of VE and VP-based models when facing the same strength guidance.

## 4 GUIDANCE FOR VE MODELS: POOR BEGINNING, FAST IMPROVEMENT

From the experiments for the reverse SDE (Fig. 1 and 7), we observe that when $\eta$ is small, VE models have a lower classification confidence compared with VP models. However, when $\eta$ becomes larger, VE models fast converge to a higher classification confidence. In this part, we explain the empirical observations. When considering the conditional generation with $\eta = 0$, we prove that the order of error term for VE models is $1/\sigma_T$, which is much larger than the $\exp(-T)$ one for VP models (Sec. 4.2). For positive $\eta$, we prove that the convergence rate of VE models is faster than VP, and VE models with reverse SDE achieve the same classification confidence with reverse PFODE (Sec. 4.3).

### 4.1 TARGET DISTRIBUTION AND CLASSIFICATION CONFIDENCE

In this work, following the setting of Wu et al. (2024), we consider a mixture of Gaussian target distribution $p_* \overset{d}{=} \sum_{y \in \mathcal{Y}} w_y \mathcal{N}(\mu_y, \Sigma)$ with each modal representing a class, where $\mathcal{Y} := \{1, 2, \ldots, |\mathcal{Y}|\}$ and $\sum_{y \in \mathcal{Y}} w_y = 1$. Under this assumption, the $s_t(x, y)$ and $\nabla_x \log c_t(x, y)$ has a close form:

$$s_t(x, y) = \nabla_x \log p_t(x|y) = -\Sigma_t^{-1} x + m_t \Sigma_t^{-1} \mu_y$$

and

$$\nabla_x \log c_t(x, y) = \nabla_x \log p_t(y \mid x) = m_t \Sigma_t^{-1} \mu_y - \sum_{y' \in \mathcal{Y}} m_t q_t(x, y') \Sigma_t^{-1} \mu_{y'},$$

where $\Sigma_t := m_t^2 \Sigma + \sigma_t^2 I_d$, and

$$q_t(x, y) := \frac{w_y \exp\left(m_t \left\langle \Sigma_t^{-1}\mu_y, x \right\rangle - m_t^2 \left\langle \mu_y, \Sigma_t^{-1}\mu_y \right\rangle /2\right)}{\sum_{y' \in \mathcal{Y}} w_{y'} \exp\left(m_t \left\langle \Sigma_t^{-1}\mu_{y'}, x \right\rangle - m_t^2 \left\langle \mu_{y'}, \Sigma_t^{-1}\mu_{y'} \right\rangle /2\right)}$$

is the posterior probability of having label $y$. In this work, we directly use the above closed form to do a clearer discussion on the influence of $\eta$ in generating the target conditional distribution. To

measure the distance between the generated samples and the target cluster, similar to Wu et al. (2024), we define the following classification confidence

$$\mathcal{P}(x, y) := q_0(x, y) = \frac{w_y \exp\left(\langle \Sigma^{-1}\mu_y, x \rangle - \langle \mu_y, \Sigma^{-1}\mu_y \rangle / 2\right)}{\sum_{y' \in \mathcal{Y}} w_{y'} \exp\left(\langle \Sigma^{-1}\mu_{y'}, x \rangle - \langle \mu_{y'}, \Sigma^{-1}\mu_{y'} \rangle / 2\right)}, \tag{3}$$

and discuss the influence of $\eta$ for the classification confidence.

## 4.2 VE MODELS WITHOUT GUIDANCE HAVE A LOWER CLASSIFICATION CONFIDENCE

As shown in Fig. 3, without guidance, the conditional VE-based diffusion models ($\eta = 0$) have a smaller classification confidence compared with VP-based models. In this part, we explain why VE-based models without guidance have a lower classification confidence. With the GMM $p_*$, the reverse PFODE (Eq. 1) has the following form for VP and VE-based models (assume our target class is $y$):

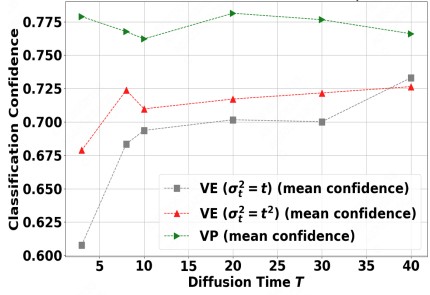

Figure 3: Results without Guidance

$$\text{VP: } \frac{dz_t}{dt} = \mu_y e^{-T+t}, \text{ VE: } \frac{dz_t}{dt} = \frac{g(T-t)(-z_t + \mu_y)}{2(1 + \sigma_{T-t}^2)}.$$

For these processes, we obtain the closed-form solution, which have different dependence on $\mu_y$.

**Theorem 4.1.** *Considering GMM $p_*$ with $\Sigma = I_d$ and reverse PFODE process without guidance (Equation (1), $\alpha = 0$). Then, for VP-based models, the closed-form solution has the following form:*

$$z(t) = z(0) + \mu_y e^{-T}\left(e^t - 1\right), \quad z(0) \sim \mathcal{N}(0, I_d).$$

*For VE-based models, the closed form solution has the following form:*

$$z(t) = \sqrt{\frac{1 + \sigma_{T-t}^2}{1 + \sigma_T^2}} z(0) + \mu_y \left[1 - \sqrt{\frac{1 + \sigma_{T-t}^2}{1 + \sigma_T^2}}\right], \quad z(0) \sim \mathcal{N}(0, \sigma_T^2 I_d).$$

Then, for the VP-based models, we know that $z^{\text{VP}}(T) \sim \mathcal{N}((1 - e^{-T})\mu_y, I_d)$. For the VE-based models, we have that $z^{\text{VE}}(T) \sim \mathcal{N}((1 - \sqrt{\frac{1}{\sigma_T^2 + 1}})\mu_y, \frac{\sigma_T^2}{1 + \sigma_T^2} I_d)$. It is clear that the $z^{\text{VE}}(T)$ is farther away from the ground truth target distribution $\mathcal{N}(\mu_y, I_d)$ compared with the VP-based models due to the $\text{Poly}(1/T)$ instead of $\exp(-T)$ of VP-based models. Hence, without any guidance, VE-based models have a lower classification confidence. For different VE-based models, the error of VE (EDM, $\sigma_t^2 = t^2$) has the order of $1/T$, which is better than the $1/\sqrt{T}$ error of VE (SMLD, $\sigma_t^2 = t$), which also matches the empirical observation Karras et al. (2022). Our simulation experiment also supports the theoretical results. As shown in Figure 3, without guidance ($\eta = 0$), the classification confidence of VP is larger than VE, and the confidence of VE ($\sigma_t^2 = t^2$) is larger than VE ($\sigma_t^2 = t$). Furthermore, when $T$ becomes larger, the error of VE ($1/\sigma_T$) becomes smaller, which leads to a higher classification confidence.

When considering the reverse SDE, there is an additional $B_t$ term in the closed-form of the VE and VP due to the Ito integral. Due to the additional randomness, under the reverse SDE setting, the classification confidence for VP and VE-based models is lower than that under the PFODE setting. However, since the Brownian motion term $B_t$ exists in both VP and VE settings, the conclusion of Theorem 4.1 still holds that the classification confidence of VE-based models is still lower than that of VP-based models under the reverse SDE, which matches our empirical observation.

## 4.3 VE MODELS ENJOYS A FAST CONVERGENCE RATE W.R.T. THE GUIDANCE STRENGTH

In the above part, we prove that without guidance, VE-based models have a lower classification confidence compared with VP-based models. However, as shown in Figure 1, when the guidance strength $\eta$ increases fast, the classification confidence of VE models increases fast and finally achieves the same confidence level as the reverse PFODE. On the contrary, the VP-based models can not achieve the same confidence with the reverse PFODE setting.

In this part, similar to previous theoretical works on the guidance diffusion Wu et al. (2024); Chidambaram et al. (2024), we do a detailed analysis on the 2-GMM case with $\Sigma = I_d$ and let $\mathcal{Y} = \{1, 2\}$. Without loss, we assume guidance is towards the cluster that has label 1 (We discuss the general GMM setting in Corollary 4.5). Then, we prove the convergence guarantee of the classification confidence w.r.t. the guidance strength $\eta$ (compared with conditional VE-based models without guidance) is at least $1 - \eta^{-1}(\log \eta)^2$, which is much faster than the one for VP-based models.

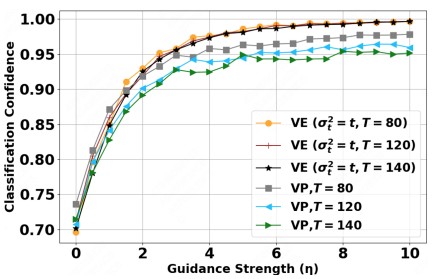

Figure 4: Results with Different $T$.

**Theorem 4.2.** *Considering* 2-*GMM* $p_*$ *with* $\Sigma = I_d$ *and reverse SDE process (Eq. 1, $\alpha = 1$), the following results hold almost surely*

*1. If $\langle x_0, \mu_1 - \mu_2 \rangle \geq \langle z_0, \mu_1 - \mu_2 \rangle$, then $\mathcal{P}(x_t, 1) \geq \mathcal{P}(z_t, 1)$ for all $t \in [0, T]$.*

*2. If $\langle x_0, \mu_1 - \mu_2 \rangle \geq \langle z_0, \mu_1 - \mu_2 \rangle$, then for all $t \in [0, T]$*

$$\mathcal{P}(x_T, 1) \geq \frac{\mathcal{P}(\bar{z}_T, 1)}{\mathcal{P}(z_T, 1) + (1 - \mathcal{P}(z_T, 1)) \cdot \exp(-\mathcal{U})} \geq \mathcal{P}(z_T, 1) \qquad (4)$$

*where $\mathcal{U}$ is any non-negative number such that*

$$\mathcal{U} \leq \frac{2}{1+T} \langle x_0 - z_0, \mu \rangle + \frac{8}{3} \left(1 - \frac{1}{(1+T)^3}\right) \eta \|\mu\|_2^2 \min\left\{\mathcal{F}\left(\max_{0 \leq t \leq T} \mathcal{P}(z_t, 1), \mathcal{U}\right), w_2\right\},$$

*with $\mu = (\mu_1 - \mu_2)/2$, $\mathcal{F}(p, u) = \frac{(1-p)e^{-u}}{p+(1-p)e^{-u}}$, and $\Delta_1 = \left|\|\mu_1\|_2^2 - \|\mu_2\|_2^2\right|$.*

*3. By setting $e^{-\mathcal{U}} = \eta^{-1}(\log \eta)^2$, the above inequality holds as $\eta$ is large enough and the convergence rate is at least $1 - O\left(\eta^{-1}(\log \eta)^2\right)$.*

Similar to Wu et al. (2024), due to the stochastic property of the reverse SDE, the result only holds almost surely. Compared with the results $1 - \eta^{-e^{-T}}(\log \eta)^{2e^{-T}}$ of VP-based models under the reverse SDE setting, it is clear that the results of Theorem 4.2 is faster and not influenced by the diffusion time $T$. Our simulation results also support the theoretical results. As shown in Figure 4, the convergence rate of VE-based models w.r.t. $\eta$ is not influenced by the diffusion time $T$. On the contrary, as $T$ becomes larger, the confidence of VP-based models becomes smaller.

With a similar proof idea, we can also prove the lower bound of the convergence guarantee for VE-based models with the reverse PFODE process, which has the same order as the reverse SDE.

**Corollary 4.3.** *Considering* 2-*GMM* $p_*$ *with and* $\Sigma = I_d$ *and reverse PFODE process (Equation (1), $\alpha = 0$). Then, if $\langle x_0, \mu_1 - \mu_2 \rangle \geq \langle z_0, \mu_1 - \mu_2 \rangle$, then for all $t \in [0, T]$*

$$\mathcal{P}(x_T, 1) \geq \frac{\mathcal{P}(z_T, 1)}{\mathcal{P}(z_T, 1) + (1 - \mathcal{P}(z_T, 1)) \cdot \exp(-\mathcal{U})} \geq \mathcal{P}(z_T, 1)$$

*where $\mathcal{U}$ is any non-negative number such that*

$$\mathcal{U} \leq \frac{2}{\sqrt{1+T}} \langle x_0 - z_0, \mu \rangle + \frac{8}{5} \eta (1 - \frac{1}{(1+T)^{2.5}}) \|\mu\|_2^2 \min\left\{\mathcal{F}\left(\max_{0 \leq t \leq T} \mathcal{P}(z_t, 1), \mathcal{U}\right), w_2\right\}.$$

*Furthermore, the convergence rate is at least $1 - O\left(\eta^{-1}(\log \eta)^2\right)$.*

Combined with Theorem 4.2, Corollary 4.3 and the results of Wu et al. (2024), we know that the convergence guarantee for VE with reverse SDE and PFODE and VP with reverse PFODE are both $1 - O\left(\eta^{-1}(\log \eta)^2\right)$, which is faster than $1 - \eta^{-e^{-T}}(\log \eta)^{2e^{-T}}$ for VP with reverse SDE. Hence, the first three settings will converge to almost the same confidence level, and the last setting will have a lower confidence level. Our experiments also support this discussion (Figure 1).

**Extension to multi-modal GMM.** In this work, we mainly focus on the 2-modal GMM to clearly explain the phenomenon of VE-based models when facing different strength guidance. Similar to Assumption 3.1 of Wu et al. (2024), we can extend our convergence guarantee analysis to the multi-modal GMM with an additional assumption on $\mu_y$.

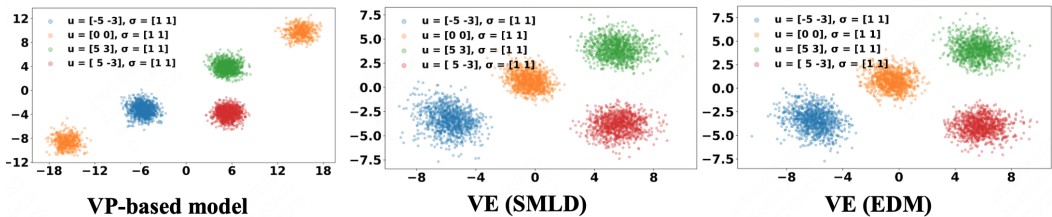

Figure 6: Influence of Guidance for VP and VE-based models (Reverse SDE, 4-GMM, $\eta = 80$).

**Assumption 4.4.** There exists $\mu_0 \in \mathbb{R}^d$ that satisfies ($y$ is our target class): (1) for $\forall y' \in \mathcal{Y}$, $|\langle \mu_y - \mu_0, \mu_{y'} - \mu_0 \rangle| \leq \epsilon$ hold for some positive constant $\epsilon$; (2) $\epsilon \leq \|\mu_y - \mu_0\|_2^2 / 3$; (3) $\Sigma = I_d$.

The above assumption indicates that the mean vectors of each cluster are almost orthogonal to one another and do not influence each other, which simplifies the analysis. With this additional assumption, for the VE-based models with reverse SDE, we can prove a $1 - \eta^{-1}(\log \eta)^2$ result, which is still faster than the one for VP-based models.

**Corollary 4.5.** *Considering $p_* = \sum_{y \in \mathcal{Y}} w_y \, \mathrm{N}(\mu_y, \Sigma)$ and reverse SDE process. Assume Assumption 4.4 holds. Let $\xi_w = 1 - w_y / (w_y + \min_{y' \neq y} w_{y'})$. Then, if $\langle x_0, \mu_y - \mu_{y'} \rangle \geq \langle z_0, \mu_y - \mu_{y'} \rangle$, then for all $t \in [0, T]$*

$$\mathcal{P}(x_T, 1) \geq \frac{\mathcal{P}(z_T, 1)}{\mathcal{P}(z_T, 1) + (1 - \mathcal{P}(z_T, 1)) \cdot \exp(-\mathcal{U})} \geq \mathcal{P}(z_T, 1)$$

*where $\mathcal{U}$ is any non-negative number such that for any $y' \neq y$*

$$\mathcal{U} \leq \frac{1}{1+T} \langle x_0 - z_0, \mu_y - \mu_{y'} \rangle$$

$$+ \frac{2}{3} \left(1 - \frac{1}{(1+T)^3}\right) \eta \min \left\{ \mathcal{F}\left(\max_{0 \leq t \leq T} \mathcal{P}(z_t, 1), \mathcal{U}\right), \xi_w \right\} \left(\|\mu_y - \mu_0\|_2^2 - 3\varepsilon\right).$$

*Furthermore, the convergence rate is at least $1 - O\left(\eta^{-1}(\log \eta)^2\right)$.*

**Influence of Variance.** In the above analysis, we provide the convergence guarantee with $\Sigma = I_d$. However, it is possible for different clusters to have different variances for real-world datasets. By conduct simulate experiments on the 2-modal GMM with different variance ($\Sigma_1 = 0.5 I_d, \Sigma_2 = I_d$, Fig. 5 and $\Sigma_1 = 2 I_d, \Sigma_2 = I_d$, Fig. 13), we show that VE-based models still have a faster convergence rate compared with VP models with reverse SDE, which indicates our theoretical guarantee should hold for more general GMM (multi-modal and different variance for each cluster).

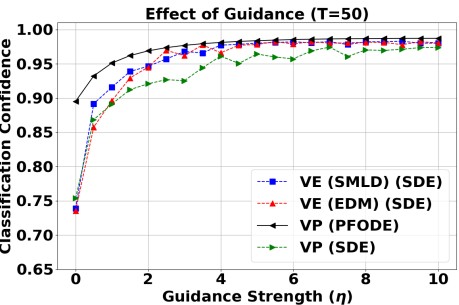

Figure 5: Results for $\Sigma_1 = 0.5 I_d, \Sigma_2 = I_d$.

For the multi-modal GMM, we have provided the guarantee in the above part. For the different variance, since the (conditional) score has a more complex closed form, we left the theoretical guarantee for GMM with different variance as an interesting and important future work.

## 5 VE MAINTAIN MULTI-MODAL PROPERTY DURING GUIDANCE PROCESS

In this part, we analyze a mode collapse example for VP-based models when facing strong guidance and intuitively explain why VE-based models can alleviate the mode collapse phenomenon. For the VP-based models, Wu et al. (2024) observe the mode collapse in a 3-modal GMM (which do not satisfy the additional assumption in Assumption 4.4):

$$p_* = \frac{1}{3} \mathcal{N}(-\mu, I_d) + \frac{1}{3} \mathcal{N}(0, I_d) + \frac{1}{3} \mathcal{N}(\mu, I_d) .$$

For each modal, Wu et al. (2024) use the guidance corresponding to it to guide diffusion models to generate samples. As shown in Figure 2 (a), when facing a large guidance and the target modal

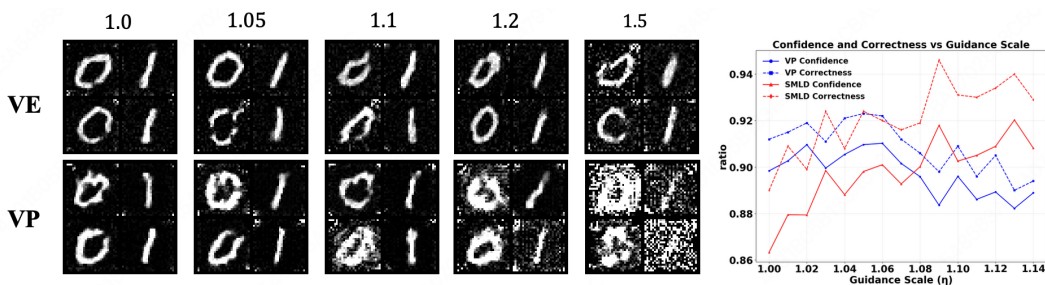

Figure 7: Result of Different Forward Processes on MNIST Dataset.

is the modal with $0$ mean, the VP-based models can not generate the target distribution, the center component tends to vanish, and the generated samples are push towards side. In other words, facing strong guidance, the center mode collapses for the VP-based models.

Our intuitive explanation is that since the VP forward process will convert each modal into $\mathcal{N}(0, I_d)$:

$$p_t = \frac{1}{3}\mathcal{N}\left(-e^{-t}\mu, I_d\right) + \frac{1}{3}\mathcal{N}\left(0, I_d\right) + \frac{1}{3}\mathcal{N}\left(e^{-t}\mu, I_d\right) ,$$

which indicates that at the end of the forward process, the three modals are almost the same and both have a $0$ mean. Then, when diffusion models reverse the process and generates models, diffusion models is hard to distinguish the center modal with $0$ mean and then is guided to the side with a strong non-zero guidance [3]. However, for the forward process of VE-based models, the mean (modal) information of the target distribution is preserved:

$$p_t = \frac{1}{3}\mathcal{N}\left(-\mu, (1 + \sigma_t^2)I_d\right) + \frac{1}{3}\mathcal{N}\left(0, (1 + \sigma_t^2)I_d\right) + \frac{1}{3}\mathcal{N}\left(\mu, (1 + \sigma_t^2)I_d\right) .$$

Then, the corresponding reverse process is more sensitive to each modal (even using pure Gaussian $\mathcal{N}(0, \sigma_T^2 I_d)$ instead of $p_T$) and can alleviate the mode collapse phenomenon. To support our intuition, we also do simulation experiments on the different VE-based models. As shown in Figure 2 (b), the VE-based models maintain the 3-modal distribution instead of ignoring the center modal when facing strong guidance. We also do more experiments beyond the 3 GMM target distribution. As shown in Figure 6, the VP-based models still suffer from the mode collapse, meanwhile the VE-based models can still generate the correct number of modals. We also conduct experiments for VP, VE (SMLD), and VE (EDM) under the reverse PFODE setting (3 GMM and 4 GMM) in Appendix A, which have similar results to Figure 2 and Figure 6.

## 6 EXPERIMENTS

In this section, we conduct experiments on the MNIST, CIFAR10 and CELEBA datasets to show that VE-based models perform better than VP-based models, which supports our theoretical guarantee. From a quantitative perspective, as shown Figure 7 and Figure 8, with a large guidance, VE-based models have a better performance and diversity. On the contrary, when facing strong guidance, VP-based models generate samples with low diversity (for example, VP models have a high probability to generate red cars) or distorted images (distorted male faces in CelebA64 experiments), which supports our results that VE has a better ability to maintain the multi-modal property.

From the classification confidence, we evaluate performance on MNIST using two quantitative metrics: (1) Confidence, measured by the average probability assigned to target labels by the pre-trained classifier; (2) Accuracy, measured by the probability of successful predictions. As shown in Figure 7, when using a small guidance scale, the confidence and accuracy have been improved for all models, which indicates the guidance-based method is helpful in conditional generation. However, as the guidance scale increased, the VE and VP-based models showed different performances. For the VP-based models, a large guidance leads to distorted images, which suffer from lower confidence and accuracy. On the contrary, the VE-based models enjoy a higher confidence level and substantially lower distortion. For the diversity, similar to Zhu et al. (2017), we evaluate on CIFAR10 with the LPIPS metric. More specifically, we generate samples by the model and calculate the LPIPS between

---

[3]We note that Wu et al. (2024) provide a precise theoretical analysis for this phase shift under the VP-based models, and this part mainly provide an intuitive discussion for VP and VE-based models.

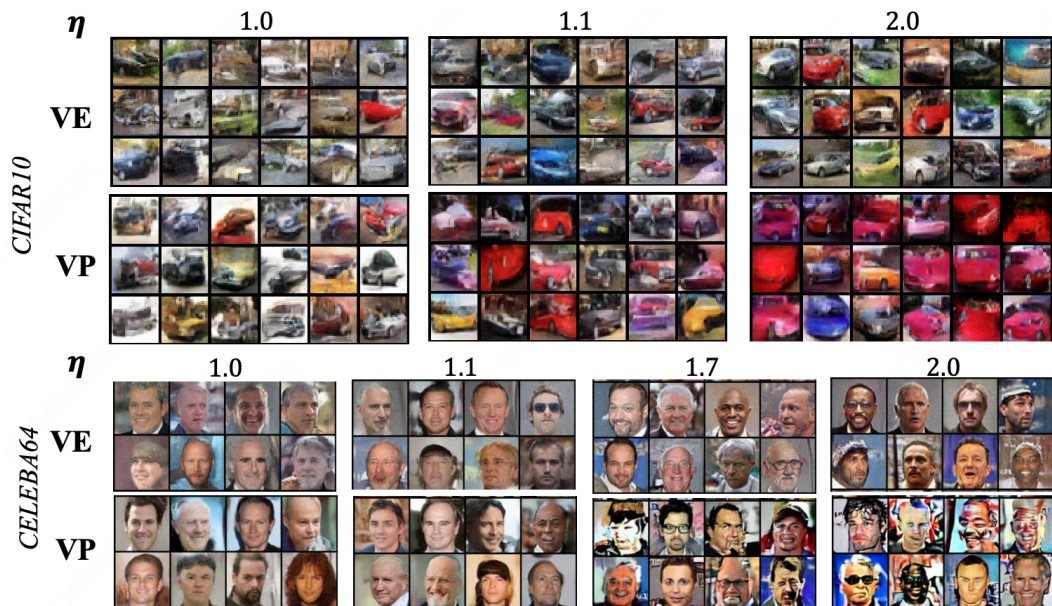

Figure 8: Experiments on CIFAR10 and CELEBA64 Dataset. For CIFAR10, the targe label is car. For CELEBA64, the target lable is male images.

samples. Then, a higher LPIPS indicates better diversity. Without any guidance, the LPIPS for VP and VE based models are $0.1777$ and $0.1772$, respectively. However, when guidance becomes larger ($\eta = 2$), the LPIPS becomes $0.1364$ for VP models, indicating that these models suffer from mode collapse. On the contrary, with ($\eta = 2$), LPIPS is $0.1731$ for VE models, which means these models maintain the multi-modal property. The above results indicate that VE-based models are more suitable for conditional generation and support our theoretical guarantee.

We note that this experiment aims to illuminate the interplay between guidance and different models, rather than to achieve state-of-the-art generation performance on complex datasets. An interesting future direction would be to extend this analysis to large-scale datasets and discuss how to design guidance methods with VE property.

## 7 CONCLUSION

In this work, we elucidate the influence of guidance for VE-based models and explain why VE (EDM) achieves SOTA performance in the conditional generation from the classification confidence and diversity perspectives. For the classification confidence, we prove that under the reverse SDE setting, the convergence rate w.r.t. the guidance strength $\eta$ is $1 - \eta^{-1}(\log \eta)^2$ for VE (EDM), which is much faster than the one $1 - \eta^{-e^{-T}}(\log \eta)^{2e^{-T}}$ of VP models and achieve the same rate with the deterministic sample process. As a result, the VE (EDM) can be more aligned with the given condition, which leads to better results in the condition generation. For the diversity, we intuitively explain why VE (EDM) can preserve the multi-modal property and VP models suffer the mode collapse phenomenon by analyzing the forward diffusion process of different models.

Combined with these two results, we show that VE (EDM) has a stronger ability in aligning the given condition and maintaining the multi-modal property from the theoretical perspective, which explains the great performance of VE (EDM) in the conditional generation task. Our theoretical results are supported by the simulation and real-world experiments.

**Future work and limitation.** In this work, we analyze the GMM distribution and explain the success of VE (EDM) in the conditional generation. Though the multi-modal property of GMM distribution is an important feature of real-world data, it is an interesting future work to analyze the role of guidance for VE-based models in a general distribution. Furthermore, when considering diversity, we provide an intuitive explanation to show why VE-based models perform better in maintaining the multi-modal data. In future work, our aim is to provide a strict theoretical guarantee for the diversity of VE-based models when facing different strength guidance. We also regard the impact of guidance on rectified flow models as interesting future work.

**Ethics statement.** Our work aims to deepen the understanding of the great performance of the guidance-based method for the conditional generation. Therefore, this work can be viewed as an important step in improving the quality of conditional generative models, and the societal impact is similar to general generative models (Mirsky and Lee, 2021).

**Reproducibility statement.** The detail and description of the real-world experiments are provided in Appendix B.2 and Appendix G. We detail the model architectures, training configurations, hyperparameters, and evaluation protocols to ensure full reproducibility.

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

APPENDIX

# A  THE USE OF LARGE LANGUAGE MODELS (LLMS)

As a theoretical work, large language models were only used for checking grammar. All analysis, experiments, writing decisions and discussion are completed entirely by the authors.

# B  ADDITIONAL EXPERIMENTS

## B.1  THE INFLUENCE OF GUIDANCE FOR REVERSE PFODE

As a supplement to Figure 1, we provide the convergence rate w.r.t. the $\eta$ under the reverse SDE and PFODE simultaneously. As shown in Figure 9, the classification confidence for the reverse PFODE (including VP, VE (SMLD) and VE (EDM)) and reverse SDE for VE-based models (VE (SMLD) and VE (EDM)) fast converge to $1$. On the contrary, the VP-based models can not achieve the same order classification confidence and are slower than other models. These results also match our theoretical results that for reverse PFODE and reverse SDE with VE forward process, the convergence guarantee is $1 - \eta^{-1}(\log \eta)^2$ . For the VP-based models with reverse SDE, the convergence guarantee is a slower one $1 - \eta^{-e^{-T}}(\log \eta)^{2e^{-T}}$ .

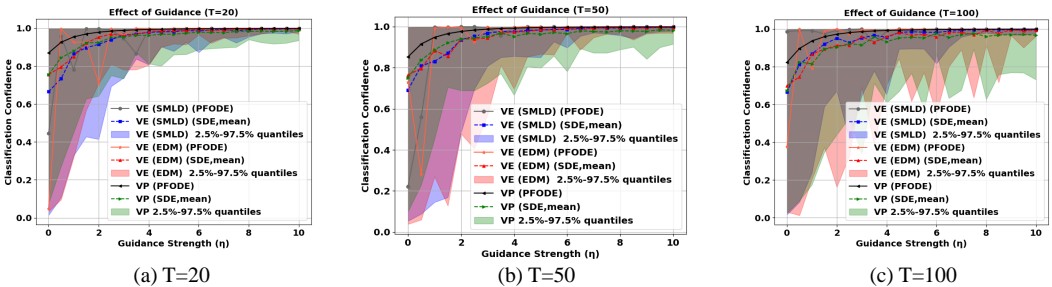

(a) T=20        (b) T=50        (c) T=100

Figure 9: Influence of Guidance for Classification confidence (VP, VE (SMLD) adn VE (EDM)).

## B.2  THE EXPERIMENTS ON THE STRONG GUIDANCE

In this part, we provide the simulation results for different diffusion models when facing strong guidance under the PFODE setting. Then, similar to the reverse SDE setting, we show that VE-based models have a strong ability to maintain the multi-modal property. On the contrary, the VP-based models suffer from modal collapse.

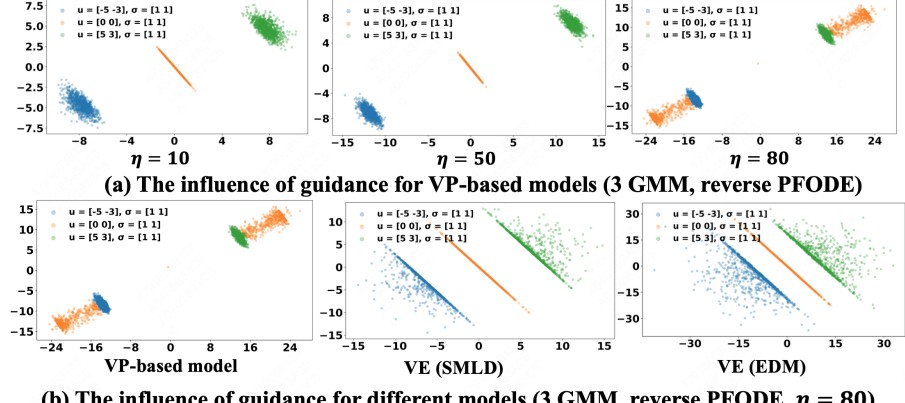

Figure 10: The Influence of Guidance for Multi-modal Property (3GMM, reverse PFODE).

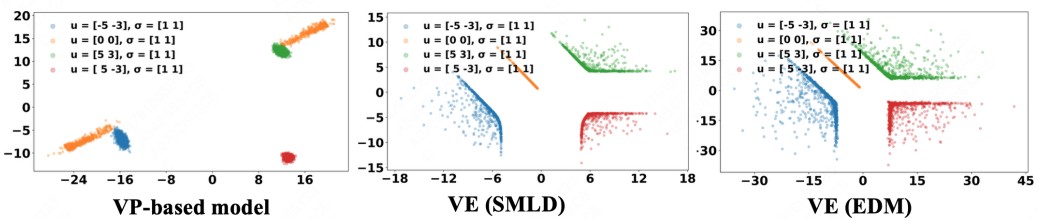

Figure 11: The Influence of Guidance for Multi-modal Property (4GMM, reverse PFODE).

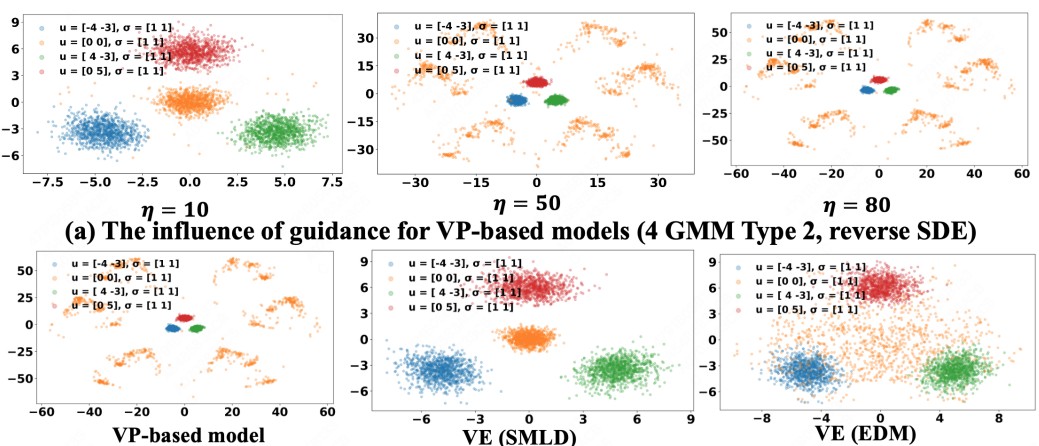

(a) The influence of guidance for VP-based models (4 GMM Type 2, reverse SDE)

(b) The influence of guidance for different models (4 GMM Type 2, reverse SDE , $\eta = 80$)

Figure 12: The Influence of Guidance for Multi-modal Property (4GMM Type 2, reverse SDE).

### B.3    THE INFLUENCE OF CLUSTER VARIANCE

Figure 13 shows that even though each cluster of GMM has a different variance, VE-based models still have a better performance compared with VP-based models, which provides some intuition that our theoretical guarantee has the potential to extend to a more general setting.

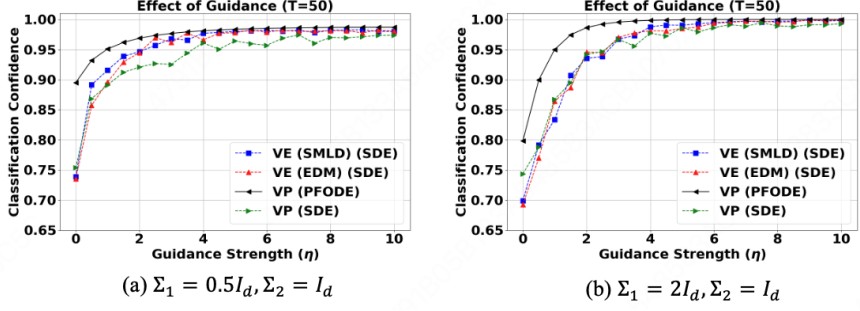

(a) $\Sigma_1 = 0.5I_d, \Sigma_2 = I_d$          (b) $\Sigma_1 = 2I_d, \Sigma_2 = I_d$

Figure 13: The Experiments with different variance.

The above experiments are conducted on a GeForce RTX 4090. For the score function, we adopt the closed-form solution of the score for the GMM target distribution. Hence, we do not need to train a neural network. For the stepsize of diffusion models in the sampling process, we adopt uniform steps with 0.1 stepsize, and the diffusion time $T$ is provided in the figures. Each experiment takes 3 minutes.

## C   THE CALCULATION OF POSTERIOR PROBABILITY FOR VE-BASED MODELS

As a starting point, we first provide an upper bound for $q_{T-t}(x, y)$ under the general diffusion process. We know that

$$q_{T-t}(x_t, y)$$

$$= \frac{w_y}{w_y + \sum_{y' \neq y} w_{y'} \exp\left(m_{T-t} \Sigma_{T-t}^{-1} \langle x_t, \mu_{y'} - \mu_y \rangle - m_{T-t}^2 \Sigma_{T-t}^{-1} \left(\|\mu_{y'}\|_2^2 - \|\mu_y\|_2^2\right)/2\right)}$$

$$= \frac{\widetilde{q}_{T-t}(x_t, y)}{\widetilde{q}_{T-t}(x_t, y) + (1 - \widetilde{q}_{T-t}(x_t, y)) \cdot \exp\left(-\left(m_{T-t}^2 - m_{T-t}\right) \Sigma_{T-t}^{-1} \left(\|\mu_{y'}\|_2^2 - \|\mu_y\|_2^2\right)/2\right)}$$

$$\leq \frac{\widetilde{q}_{T-t}(x_t, y)}{\widetilde{q}_{T-t}(x_t, y) + (1 - \widetilde{q}_{T-t}(x_t, y)) \cdot \exp\left(-C(\Delta, m_T, m_0, \Sigma_0, \Sigma_T)\right)},$$

where

$$\widetilde{q}_{T-t}(x_t, y) = \frac{w_y}{w_y + \sum_{y' \neq y} w_{y'} \exp\left(m_{T-t} \Sigma_{T-t}^{-1} \langle x_t, \mu_{y'} - \mu_y \rangle - m_{T-t} \Sigma_{T-t}^{-1} \left(\|\mu_{y'}\|_2^2 - \|\mu_y\|_2^2\right)/2\right)},$$

and $C(\Delta, m_T, m_0, \Sigma_0, \Sigma_T)$ is a constant depends on $\Delta$ and the forward process.

For the VE-based diffusion models, since $m_t = 1$, we have the following inequality

$$q_{T-t}(x_t, y) \leq \frac{\widetilde{q}_{T-t}(x_t, y)}{\widetilde{q}_{T-t}(x_t, y) + (1 - \widetilde{q}_{T-t}(x_t, y))}$$

For the VP-based diffusion models, we have that

$$q_{T-t}(x_t, y) \leq \frac{\widetilde{q}_{T-t}(x_t, y)}{\widetilde{q}_{T-t}(x_t, y) + (1 - \widetilde{q}_{T-t}(x_t, y)) \cdot \exp(-\Delta/(8 \max\{\sigma^2, 1\}))}.$$

Hence $C(\Delta, m_T, m_0, \Sigma_0, \Sigma_T) = 0$ for the VE-based models and is equal to $\Delta/(8 \max\{\sigma^2, 1\})$ for VP-based models. In the following process, without ambiguity, we will abbreviate $C(\Delta, m_T, m_0, \Sigma_0, \Sigma_T)$ to $C$.

If $\exp\left(\langle x_t, \mu_y \rangle - \|\mu_y\|_2^2/2\right) = \max_{y' \in \mathcal{Y}} \exp\left(\langle x_t, \mu_{y'} \rangle - \|\mu_{y'}\|_2^2/2\right)$, then one can verify that

$$\widetilde{q}_{T-t}(x_t, y) = \frac{w_y \exp\left(m_{T-t} \Sigma_{T-t}^{-1} \langle x_t, \mu_y \rangle - m_{T-t}^2 \Sigma_{T-t}^{-1} \|\mu_y\|_2^2/2\right)}{\sum_{y' \in \mathcal{Y}} w_{y'} \exp\left(m_{T-t} \Sigma_{T-t}^{-1} \langle x_t, \mu_{y'} \rangle - m_{T-t}^2 \Sigma_{T-t}^{-1} \|\mu_{y'}\|_2^2/2\right)} \leq \mathcal{P}(x_t, y) .$$

On the other hand, if $\exp\left(\langle x_t, \mu_y \rangle - \|\mu_y\|_2^2/2\right) \neq \max_{y' \in \mathcal{Y}} \exp\left(\langle x_t, \mu_{y'} \rangle - \|\mu_{y'}\|_2^2/2\right)$, we know that $\widetilde{q}_{T-t}(x_t, y) \leq w_y/(w_y + \min_{y' \neq y} w_{y'})$, which indicates

$$q_{T-t}(x_t, y) \leq \frac{w_y}{w_y + \min_{y' \neq y} w_{y'} \exp(-C)} .$$

Combined with these two situations, we have the following bound for $q_{T-t}(x_t, y)$:

$$q_{T-t}(x_t, y) \leq \max\left\{G\left(\mathcal{P}(x_t, y)\right), G\left(w_y/\left(w_y + \min_{y' \neq y} w_{y'}\right)\right)\right\},$$

where $G(x) := x/(x + (1 - x) \cdot \exp(-C))$ is a function that maps $[0, 1]$ to $[0, 1]$ (with the definition of $C$ for the VP and VE-based models). We note that for $\forall x \in [0, 1]$,

$$G'(x) = \frac{\exp(-C)}{[x + (1 - x) \cdot \exp(-C)]^2} \in [\exp(-C), \exp(C)] .$$

Let $\xi_w := 1 - w_y/(w_y + \min_{y' \neq y} w_{y'}) > 0$. We note that $G(1) = 1$, which indicates $1 - G\left(\mathcal{P}(x_t, y)\right) \geq \exp(-C) \cdot (1 - \mathcal{P}(x_t, y))$ and $1 - G(1 - \xi_w) \geq \exp(-C) \cdot \xi_w$.

When considering 2-modal GMM setting (used in Theorem 4.2 and Corollary 4.3) , the above results is simplified to

$$1 - G\left(\mathcal{P}(x_t, 1)\right) \geq e^{-C}(1 - \mathcal{P}(x_t, 1)), \quad 1 - G(w_1) \geq e^{-C}(1 - w_1) .$$

Then, we know that

$$1 - q_{T-t}(x_t, 1) \geq e^{-C} \min\{1 - \mathcal{P}(x_t, 1), 1 - w_1\} \tag{5}$$

# D CLASSIFICATION CONFIDENCE CONVERGENCE GUARANTEE

**Theorem 4.2.** *Considering 2-GMM $p_*$ with $\Sigma = I_d$ and reverse SDE process (Eq. 1, $\alpha = 1$), the following results hold almost surely*

*1. If $\langle x_0, \mu_1 - \mu_2 \rangle \geq \langle z_0, \mu_1 - \mu_2 \rangle$, then $\mathcal{P}(x_t, 1) \geq \mathcal{P}(z_t, 1)$ for all $t \in [0, T]$.*

*2. If $\langle x_0, \mu_1 - \mu_2 \rangle \geq \langle z_0, \mu_1 - \mu_2 \rangle$, then for all $t \in [0, T]$*

$$\mathcal{P}(x_T, 1) \geq \frac{\mathcal{P}(\bar{z}_T, 1)}{\mathcal{P}(z_T, 1) + (1 - \mathcal{P}(z_T, 1)) \cdot \exp(-\mathcal{U})} \geq \mathcal{P}(z_T, 1) \tag{4}$$

*where $\mathcal{U}$ is any non-negative number such that*

$$\mathcal{U} \leq \frac{2}{1+T} \langle x_0 - z_0, \mu \rangle + \frac{8}{3} \left(1 - \frac{1}{(1+T)^3}\right) \eta \|\mu\|_2^2 \min \left\{ \mathcal{F}\left( \max_{0 \leq t \leq T} \mathcal{P}(z_t, 1), \mathcal{U} \right), w_2 \right\},$$

*with $\mu = (\mu_1 - \mu_2)/2$, $\mathcal{F}(p, u) = \frac{(1-p)e^{-u}}{p + (1-p)e^{-u}}$, and $\Delta_1 = \left| \|\mu_1\|_2^2 - \|\mu_2\|_2^2 \right|$.*

*3. By setting $e^{-\mathcal{U}} = \eta^{-1}(\log \eta)^2$, the above inequality holds as $\eta$ is large enough and the convergence rate is at least $1 - O(\eta^{-1}(\log \eta)^2)$.*

**Proof.** Set $\mu_0 = (\mu_1 + \mu_2)/2$ and $\mu = \mu_1 - \mu_0$. Then, we have the following SDE for the VE-based models ($\sigma_t^2 = t^2$) with guidance and without guidance

$$2\,\mathrm{d}\langle x_t, \mu \rangle = \left[ g(T-t)^2 \Sigma_{T-t}^{-1} \left( -2\langle x_t, \mu \rangle + 2\|\mu_1\|_2^2 - 2\langle \mu_1, \mu_2 \rangle + 8\eta(1 - q_{T-t}(x_t, 1))\|\mu\|_2^2 \right) \right] \mathrm{d}t \\ + 2g(T-t)\langle \mathrm{d}B_t, \mu \rangle, \tag{6}$$

and

$$2\,\mathrm{d}\langle z_t, \mu \rangle = \left[ g(T-t)^2 \Sigma_{T-t}^{-1} \left( -2\langle z_t, \mu \rangle + 2\|\mu_1\|_2^2 - 2\langle \mu_1, \mu_2 \rangle \right) \right] \mathrm{d}t + 2g(T-t)\langle \mathrm{d}B_t, \mu \rangle.$$

Then, for the first part of Theorem 4.2, we can directly use the SDE comparison lemma to obtian similar results.

For the second part, with Equation (5), we know that

$$1 - q_{T-t}(x_t, 1) \geq e^{-C} \min\{1 - \mathcal{P}(x_t, 1), 1 - w_1\}.$$

with $C = 0$. With this result, we know that

$$2\,\mathrm{d}\langle x_t - z_t, \mu \rangle \geq \left[ -2g(T-t)^2 \Sigma_{T-t}^{-1} \langle x_t - z_t, \mu \rangle + 8g(T-t)^2 \Sigma_{T-t}^{-1} \eta \|\mu\|_2^2 \min\{1 - \mathcal{P}(x_t, 1), w_2\} \right] \mathrm{d}t.$$

To use the integrating factor method, we multiply $\exp\left( \int g(T-t)^2 \Sigma_{T-t}^{-1} \mathrm{d}t \right)$ on the both set. For VE-based models with $\sigma_t^2 = t$, we know that $\int g(T-t)^2 \Sigma_{T-t}^{-1} \mathrm{d}t = \int \frac{1}{\sigma^2 + (T-t)} \mathrm{d}t = -\ln \sigma^2 + (T-t)$. Then, we know that

$$\mathrm{d}\left\langle \frac{2}{\sigma^2 + (T-t)} x_t - z_t, \mu \right\rangle \geq \left[ 8\left( \frac{1}{\sigma^2 + (T-t)} \right)^2 \eta \|\mu\|_2^2 \min\{1 - \mathcal{P}(x_t, 1), w_2\} \right] \mathrm{d}.$$

Since by assumption $\langle x_0 - z_0, \mu \rangle \geq 0$, we then conclude that almost surely we have $\langle x_t - z_t, \mu \rangle \geq 0$ for all $t \in [0, T]$. If we assume $\frac{2}{\sigma^2 + (T-t)} \langle x_t - z_t, \mu \rangle \in [0, \mathcal{U}]$ for all $t \in [0, T]$, then it holds that $2\langle x_t - z_t, \mu \rangle \leq \sigma^2 \mathcal{U}$ for all $t \in [0, T]$ (Here $\sigma^2 = 1$). Then, we know that

$$1 - \mathcal{P}(x_t, 1) \geq \mathcal{F}\left( \max_{0 \leq t \leq T} \mathcal{P}(z_t, 1), \mathcal{U}\sigma^2 \right).$$

Then, we know that

$$\mathcal{U} \geq \frac{2}{\sigma^2 + T} \langle x_0 - z_0, \mu \rangle + \frac{8}{3} \left( \frac{1}{\sigma^6} - \frac{1}{(\sigma^2 + T)^3} \right) \eta \|\mu\|_2^2 \min \left\{ \mathcal{F}\left( \max_{0 \leq t \leq T} \mathcal{P}(z_t, 1), \mathcal{U}\sigma^2 \right), w_2 \right\}. \tag{7}$$

If the above inequality is not satisfied, then we know that for such $\mathcal{U}$ we have $2\langle x_T - z_T, \mu \rangle \geq \sigma^2 \mathcal{U}$ and

$$\mathcal{P}(x_T, 1) \geq \frac{\mathcal{P}(z_T, 1)}{\mathcal{P}(z_T, 1) + (1 - \mathcal{P}(z_T, 1)) \cdot \exp(-\sigma^2 \mathcal{U})}.$$

**The Convergence Guarantee for $\eta$.** In this part, we prove the convergence rate of $\eta$. We set $e^{-\mathcal{U}} = \eta^{-1}(\log\eta)^2$. Then, we know that the left hand of Equation (7) is the order of $O(\log\eta)$ and the right hand of Equation (7) is order of $O\left(\eta \wedge (\log\eta)^2\right)$. Hence, for large enough $\eta$, the inequality 7 does not hold. Plugging such $\mathcal{U}$ into Equation (7), we deduce that $\mathcal{P}(x_T, 1) \geq 1 - O\left(\eta^{-1}(\log\eta)^2\right)$ as $\eta \to \infty$. Then, we complete the proof. We note that in this part, we use VE (SMLD) with $\sigma_t^2 = t$ as an example to provide the convergence guarantee. For the VE (EDM), the proof process is exactly the same. ∎

**Proof Process of Corollary 4.3.** The proof under the PFODE setting is also the same with Theorem 4.2. For the first part, we use the ODE comparison lemma instead of the SDE comparison lemma (Hence, the results of PFODE is with probability 1 instead of almost surely). Then, since Equation (5) holds for the reverse SDE and PFODE setting at the same time. Then, we also use the integrating factor method and the following calculation to complete the proof.

**Corollary D.1.** *Considering $p_* = \sum_{y\in\mathcal{Y}} w_y\, \mathrm{N}(\mu_y, \Sigma)$ and reverse SDE process. Assume Assumption 4.4 holds. Let $\xi_w = 1 - w_y/\left(w_y + \min_{y'\neq y} w_{y'}\right)$. Then, if $\langle x_0, \mu_y - \mu_{y'}\rangle \geq \langle z_0, \mu_y - \mu_{y'}\rangle$, then for all $t \in [0, T]$*

$$\mathcal{P}(x_T, 1) \geq \frac{\mathcal{P}(z_T, 1)}{\mathcal{P}(z_T, 1) + (1 - \mathcal{P}(z_T, 1)) \cdot \exp(-\mathcal{U})} \geq \mathcal{P}(z_T, 1)$$

*where $\mathcal{U}$ is any non-negative number such that for any $y' \neq y$*

$$\mathcal{U} \leq \frac{1}{1+T}\langle x_0 - z_0, \mu_y - \mu_{y'}\rangle$$

$$+ \frac{2}{3}\left(1 - \frac{1}{(1+T)^3}\right)\eta \min\left\{\mathcal{F}\left(\max_{0 \leq t \leq T} \mathcal{P}(z_t, 1), \mathcal{U}\right), \xi_w\right\}\left(\|\mu_y - \mu_0\|_2^2 - 3\varepsilon\right).$$

*Furthermore, the convergence rate is at least $1 - O\left(\eta^{-1}(\log\eta)^2\right)$.*

**Proof.** Since our calculation for the posterior probability is based on the general GMM, we only calculate the lower bound of $\langle x_t, \mu_y - \mu_{y'}\rangle$ with Assumption 4.4. We note that the following calculation mainly following the process of Eq. (A.1) of Wu et al. (2024) and we extend the calculation to the VE setting.

$$\mathrm{d}\langle x_t, \mu_y - \mu_{y'}\rangle$$

$$= \left[g(T-t)^2\Sigma_{T-t}^{-1}\left(-\langle x_t, \mu_y - \mu_{y'}\rangle + (1 + \eta - \eta q_{T-t}(x_t, y))\|\mu_y\|_2^2 - \eta\sum_{y''\neq y} q_{T-t}(x_t, y'')\langle\mu_y, \mu_{y''}\rangle\right.\right.$$

$$\left.\left. - (1 + \eta - \eta q_{T-t}(x_t, y))\langle\mu_y, \mu_{y'}\rangle + \eta\sum_{y''\neq y} q_{T-t}(x_t, y'')\langle\mu_{y'}, \mu_{y''}\rangle\right)\right]\mathrm{d}t$$

$$+ \sqrt{2}g(T-t)\langle\mathrm{d}B_t, \mu_y - \mu_{y'}\rangle$$

Let

$$\alpha_t := g(T-t)^2\Sigma_{T-t}^{-1}, \qquad q_y := q_{T-t}(x_t, y), \qquad q_{y''} := q_{T-t}(x_t, y'').$$

Then, we have that

$$\mathrm{d}\langle x_t, \mu_y - \mu_{y'}\rangle = \left[\alpha_t\left(-\langle x_t, \mu_y - \mu_{y'}\rangle + (1 + \eta - \eta q_y)\|\mu_y\|^2 - \eta\sum_{y''\neq y} q_{y''}\langle\mu_y, \mu_{y''}\rangle\right.\right.$$

$$\left.\left. - (1 + \eta - \eta q_y)\langle\mu_y, \mu_{y'}\rangle + \eta\sum_{y''\neq y} q_{y''}\langle\mu_{y'}, \mu_{y''}\rangle\right)\right]\mathrm{d}t$$

$$+ \sqrt{2}\,g(T-t)\langle\mathrm{d}B_t, \mu_y - \mu_{y'}\rangle, \tag{8}$$

We separate in Equation (8) the $\eta$–dependent part of the drift. Define

$$G_t := (1 - q_y)\|\mu_y\|^2 - \sum_{y''\neq y} q_{y''}\langle\mu_y, \mu_{y''}\rangle - (1 - q_y)\langle\mu_y, \mu_{y'}\rangle + \sum_{y''\neq y} q_{y''}\langle\mu_{y'}, \mu_{y''}\rangle. \tag{9}$$

For $G_t$, we have that

$$G_t = (1 - q_y)\|\mu_y - \mu_0\|^2 + q_{y'}\|\mu_{y'} - \mu_0\|^2 + R_t,$$

where the error term $R_t$ is given explicitly by

$$R_t := -(1 - q_y)\langle \mu_y - \mu_0, \mu_{y'} - \mu_0 \rangle - \sum_{y'' \neq y} q_{y''}\langle \mu_y - \mu_0, \mu_{y''} - \mu_0 \rangle + \sum_{y'' \neq y} q_{y''}\langle \mu_{y'} - \mu_0, \mu_{y''} - \mu_0 \rangle.$$

(10)

Inserting Equation (9) and Equation (10) into Equation (8), we can define the error term $\bar{\mathcal{E}}_t$

$$\bar{\mathcal{E}}_t := \alpha_t \, \eta \, R_t = g(T - t)^2 \Sigma_{T-t}^{-1} \eta R_t.$$

By Assumption 4.4, for all $u, v \in \mathcal{Y}$,

$$\left| \langle \mu_u - \mu_0, \mu_v - \mu_0 \rangle \right| \leq \varepsilon.$$

Moreover,

$$\sum_{y'' \neq y} q_{y''} = 1 - q_y.$$

Hence each term in $R_t$ is bounded by

$$|(1 - q_y)\langle \mu_y - \mu_0, \mu_{y'} - \mu_0 \rangle| \leq (1 - q_y)\varepsilon,$$

$$\left| \sum_{y'' \neq y} q_{y''}\langle \mu_y - \mu_0, \mu_{y''} - \mu_0 \rangle \right| \leq (1 - q_y)\varepsilon,$$

$$\left| \sum_{y'' \neq y} q_{y''}\langle \mu_{y'} - \mu_0, \mu_{y''} - \mu_0 \rangle \right| \leq (1 - q_y)\varepsilon.$$

Therefore,

$$|R_t| \leq 3(1 - q_y)\varepsilon.$$

Consequently, the error term satisfies

$$|\bar{\mathcal{E}}_t| \leq 3\eta \, g(T - t)^2 \Sigma_{T-t}^{-1} \big(1 - q_{T-t}(x_t, y)\big)\varepsilon.$$

For VE (SMLD), we have that $g(T - t)^2 \Sigma_{T-t}^{-1} = \frac{1}{1+(T-t)} \leq 1$ (with $\Sigma = 1$). For VE (EDM), we have that $g(T - t)^2 \Sigma_{T-t}^{-1} = \frac{2(T-t)}{1+(T-t)^2} \leq 1$. Then, we have the following bound for $\bar{\mathcal{E}}_t$

$$|\bar{\mathcal{E}}_t| \leq 6\eta(1 - q_{T-t}(x_t, y))\varepsilon.$$

As a result, we have the following inequality:

$$\begin{aligned}
&\mathrm{d}\langle x_t, \mu_y - \mu_{y'} \rangle \\
&= \Big[\alpha_t\Big( -\langle x_t, \mu_y - \mu_{y'} \rangle + \|\mu_y\|_2^2 - \langle \mu_y, \mu_{y'} \rangle \\
&\qquad\qquad + \eta(1 - q_y)\|\mu_y - \mu_0\|_2^2 + \eta q_{y'}\|\mu_{y'} - \mu_0\|_2^2 \Big) + \bar{\mathcal{E}}_t\Big]\mathrm{d}t + \sqrt{2}\,g(T - t)\,\langle \mathrm{d}B_t, \mu_y - \mu_{y'} \rangle \\
&\geq \Big[\alpha_t\Big( -\langle x_t, \mu_y - \mu_{y'} \rangle + \|\mu_y\|_2^2 - \langle \mu_y, \mu_{y'} \rangle \\
&\qquad\qquad + \eta(1 - q_y)\big(\|\mu_y - \mu_0\|_2^2 - 3\varepsilon\big)\Big)\Big]\mathrm{d}t + \sqrt{2}\,g(T - t)\,\langle \mathrm{d}B_t, \mu_y - \mu_{y'} \rangle.
\end{aligned}$$

The above bounds have almost the same form compared with Equation (6), and we can bound $\mathrm{d}\langle x_t - z_t, \mu_y - \mu_{y'} \rangle$ with the same process of Theorem 4.2. Then, we complete the proof. $\blacksquare$

## E  RESULTS FOR CONDITIONAL DIFFUSION MODELS WITHOUT GUIDANCE

In this part, by showing the closed-form solution of conditional diffusion models (PFODE and SDE setting), we explain the difference performance of different diffusion models without guidance.

**Theorem 4.1.** *Considering GMM $p_*$ with $\Sigma = I_d$ and reverse PFODE process without guidance (Equation (1), $\alpha = 0$). Then, for VP-based models, the closed-form solution has the following form:*

$$z(t) = z(0) + \mu_y e^{-T} \left(e^t - 1\right), \quad z(0) \sim \mathcal{N}(0, I_d).$$

*For VE-based models, the closed form solution has the following form:*

$$z(t) = \sqrt{\frac{1+\sigma_{T-t}^2}{1+\sigma_T^2}} z(0) + \mu_y \left[1 - \sqrt{\frac{1+\sigma_{T-t}^2}{1+\sigma_T^2}}\right], \quad z(0) \sim \mathcal{N}(0, \sigma_T^2 I_d).$$

**Proof.** We know that given a target label $y$, the PFODE for the VP-based models has the following form

$$\frac{dz_t}{dt} = \mu_y e^{-T+t}.$$

Then, integrating for $t = 0$ to $t = T$, we have the following results:

$$z(t) = x(0) + \int_0^t \mu_y e^{-T+s} ds$$

$$= z(0) + \mu_y \int_0^t e^{-T+s} ds$$

$$= zx(0) + \mu_y e^{-T} \left(e^t - 1\right), \quad z(0) \sim \mathcal{N}(0, 1),$$

For the VE (EDM), we know the PFODE has the following form:

$$\frac{dz_t}{dt} = \frac{T-t}{1 + (T-t)^2} \left(-z_t + \mu_y\right).$$

Let

$$p(t) = \frac{T-t}{1 + (T-t)^2}, \quad q(t) = \frac{T-t}{1 + (T-t)^2} \mu_y.$$

Then, the PFODE for VE (EDM) is a standard linear ODE:

$$\frac{dz}{dt} + p(t)z(t) = q(t).$$

Following the standard process of solving linear ODE, we calculate

$$\mu(t) = \exp\left(\int p(t) dt\right) = \exp\left(\int \frac{T-t}{1 + (T-t)^2} dt\right).$$

Set $u = T - t$, so $du = -dt$. Then

$$\int \frac{T-t}{1 + (T-t)^2} dt = -\int \frac{u}{1 + u^2} du = -\frac{1}{2} \ln\left(1 + u^2\right) = -\frac{1}{2} \ln\left(1 + (T-t)^2\right)$$

Hence

$$\mu(t) = \left(1 + (T-t)^2\right)^{-1/2}.$$

Multiplying through by $\mu(t)$ gives

$$\left(1 + (T-t)^2\right)^{-1/2} \frac{dz_t}{dt} + \frac{T-t}{(1 + (T-t)^2)^{3/2}} z_t = \mu_y \frac{T-t}{(1 + (T-t)^2)^{3/2}}$$

One checks by the product rule that the left-hand side is

$$
\frac{d}{dt}\left[\left(1+(T-t)^2\right)^{-1/2}z_t\right]
$$

So the ODE becomes

$$
\frac{d}{dt}\left[\left(1+(T-t)^2\right)^{-1/2}z_t\right]=\mu_y\frac{T-t}{\left(1+(T-t)^2\right)^{3/2}}\,.
$$

Then, we integrate from $0$ to $t$ for the both sides:

$$
\left(1+(T-t)^2\right)^{-1/2}x(t)-\left(1+T^2\right)^{-1/2}x(0)=\mu_y\int_0^t\frac{T-s}{\left(1+(T-s)^2\right)^{3/2}}ds\,.
$$

For the integral of the right side, we know that

$$
\int_0^t\frac{T-s}{\left(1+(T-s)^2\right)^{3/2}}ds=\frac{1}{\sqrt{1+(T-t)^2}}-\frac{1}{\sqrt{1+T^2}}\,.
$$

As a result, we know that

$$
\left(1+(T-t)^2\right)^{-1/2}x(t)-\left(1+T^2\right)^{-1/2}x(0)=\mu_y\left[\frac{1}{\sqrt{1+(T-t)^2}}-\frac{1}{\sqrt{1+T^2}}\right],
$$

which indicates that

$$
x(t)=\sqrt{\frac{1+(T-t)^2}{1+T^2}}\,x(0)+\mu_y\left[1-\sqrt{\frac{1+(T-t)^2}{1+T^2}}\right]\,.
$$

$\blacksquare$

Then, the proof for VE (EDM) with the reverse PFODE is finished. For the proof for VE (SMLD) with PFODE is almost the same with

$$
p(t)=\frac{1}{2(1+T-t)},\quad q(t)=\frac{\mu_y}{2(1+T-t)}
$$

and standard solving process of linear ODE.

## F    USEFUL LEMMA

To prove the first part of Theorem 4.2 and Corollary 4.3, we directly use the ODE and SDE comparison lemma provided by Wu et al. (2024). For completeness, we provide these two lemmas in the following part (Lemma 3.4 and Lemma A.1 of Wu et al. (2024)).

**Lemma F.1** (ODE comparison lemma). *Suppose $f(t,u)$ is continuous in $(t,u)$ and Lipschitz continuous in $u$. Suppose $u(t),v(t)$ are $C^1$ for $t\in[0,T]$, and satisfy*

$$
u'(t)\le f(t,u(t)),\quad v'(t)=f(t,v(t))
$$

*In addition, we assume $u(0)\le v(0)$. Then $u(t)\le v(t)$ for all $t\in[0,T]$.*

**Lemma F.2** (SDE Comparison Lemma). *Consider the following two $m$-dimensional SDEs defined on $[0,T]$ :*

$$
X_t^1=x^1+\int_0^t b_1\left(s,X_s^1\right)\mathrm{d}s+\int_0^t\sigma_1\left(s,X_s^1\right)\mathrm{d}W_s
$$

$$
X_t^2=x^2+\int_0^t b_2\left(s,X_s^2\right)\mathrm{d}s+\int_0^t\sigma_2\left(s,X_s^2\right)\mathrm{d}W_s
$$

*We assume the following conditions: 1. $b(t, x), \sigma(t, x)$ are continuous in $(t, x)$, 2. There exists a sufficiently large constant $\mu > 0$, such that for all $x, x' \in \mathbb{R}^m$ and $t \in [0, T]$, it holds that*

$$\|b(t, x) - b(t, x')\|_2 + \|\sigma(t, x) - \sigma(t, x')\|_2 \leq \mu \|x - x'\|_2$$
$$\|b(t, x)\|_2 + \|\sigma(t, x)\|_2 \leq \mu (1 + \|x\|_2)$$

*Then the following are equivalent:*

*(i) For any $t \in [0, T]$ and $x^1, x^2 \in \mathbb{R}^m$ such that $x^1 \geq x^2$, almost surely we have $X_t^1 \geq X_t^2$ for all $t \in [0, T]$.*

*(ii) $\sigma^1 \equiv \sigma^2$, and for any $t \in [0, T], k = 1, 2, \cdots, m$,*

$$\begin{cases} (a) & \sigma_k^1 \text{ depends only on } x_k \\ (b) & \text{for all } x', \delta^k x \in \mathbb{R}^m, \text{ such that } \delta^k x \geq 0, \left(\delta^k x\right)_k = 0, \\ & b_k^1 \left(t, \delta^k x + x'\right) \geq b_k^2 \left(t, x'\right) \end{cases}$$

## G  MINIST EXPERIMENTS

This appendix provides a comprehensive description of the experimental setup and additional results that support the findings presented in the main text regarding the superior performance of the VE framework over VP under low guidance strength on the MNIST dataset. We detail the model architectures, training configurations, hyperparameters, and evaluation protocols to ensure full reproducibility.

**Dataset, Network Architecture, Training Configuration.**

- MNIST. We used the standard MNIST dataset, which consists of $60,000$ training and $10,000$ test images.
- CIFAR10. We use standard CIFAR10 dataset.
- CelebA64. We collect $10k$ image for female faces and $10k$ image for

All models shared a common U-Net backbone featuring an encoder-decoder structure with skip connections. The network was conditioned on the time step t via Gaussian Fourier feature embedding. Complete architectural details are elaborated in Appendix G.1. All models were trained from scratch for a fixed number of epochs. The optimizer is Adam, the learning rate is $1e - 4$, the batch size is 32, and the training epochs are 30.

**SDE Configuration.**   We implemented both Variance Preserving (VP) and Variance Exploding (VE) SDEs as defined by Song et al. (2020).

For VP-based models, the forward SDE is defined by (here $x$ is our $z^{\rightarrow}$)

$$dx = -0.5\beta(t)xdt + \sqrt{\beta(t)}dB_t \,,$$

and $\beta(t) = (\beta_0 + t(\beta_1 - \beta_0))^2$ where $\beta(t)$ is a linearly increasing schedule from $\beta_0 = 0.1$ to $\beta_1 = 20.0$ over the course of the diffusion process.

For VE (SMLD), the SDE forward process is defined by: $dx = \sigma^t dB_t$ where the noise schedule $\sigma$ is set to $\sigma = 15.0$.

**Sampling/Inference Configuration.**   The number of sampling steps was set to $500$ for all experiments to ensure a high-quality generation. The guidance scale ($\eta$) was swept across a logarithmic scale: $[1.0, 1.05, 1.1, 1.2, 1.3, 1.4, 1.5]$. All metrics were computed using a CNN classifier. For each experiment, metrics were calculated over a set of $10,000$ generated images to ensure statistical significance.

Note that in our implementation, we adopted a simplified training process. Instead of training a single neural network that learns $\nabla p(x|y)$ and $\nabla p(x)$ , we train a separate neural network for no classifier

situation and each target class $y$. Given the small size of the MNIST dataset and the relatively low computational cost of training these models, this simplification is feasible. It significantly simplifies the training pipeline by avoiding the need for a jointly trained classifier and the associated gradient calculations during training, allowing us to focus our analysis purely on the sampling dynamics. We acknowledge that this strategy does not scale to complex datasets due to its linear growth in computational cost during training. However, the focused comparative study presented here provides a clean and interpretable experimental framework. The insights gained are expected to generalize to the more scalable single-model conditional setting.

## G.1 DIFFUSION MODEL ARCHITECTURES

**Diffusion Model Architectures.** A single U-Net architecture is used to parameterize the score function for both VP and VE frameworks. The NN contains 4 down-sampling blocks and 4 up-sampling blocks. The down-sampling Channel is $[32, 64, 128, 256]$ and the up-sampling channel is $[256, 128, 64, 32]$.

**Classifier Architecture.** The pre-trained classifier used for all evaluation metrics was a convolutional neural network. This classifier was trained on the official MNIST training set ($60,000$ images) for 10 epochs using the Adam optimizer (learning rate $1e-4$) and cross-entropy loss. It achieved a final accuracy of 98% on the official MNIST test set ($10,000$ images), confirming its competence as an evaluator.

