# OpenReview forum: "Elucidating Guidance in Variance Exploding Diffusion Models: Fast Convergence and Better Diversity"
_ICLR.cc/2026/Conference — Submitted to ICLR 2026_

### Official Review · Reviewer_c7XG · 2025-10-29

**Soundness:** 3
**Presentation:** 3
**Contribution:** 3
**Rating:** 6
**Confidence:** 3

**Summary:**

This article studies how classifier guidance affects classification confidence and mode collapse in Variance-Exploding (VE) and Variance-Preserving (VP) SDEs. The authors derive convergence rates for classification confidence, showing it scales as $1 - \eta^{-1} (\log \eta)^{2}$ for VE-SDEs and $1 - \eta^{-e^{-T}} (\log \eta)^{2 e^{-T}}$ for VP-SDEs, where $\eta$ is the guidance strength and $T$ is the diffusion time. They also analyze the distinct mode collapse behaviors in these two SDE classes. Theoretical claims are verified through numerical experiments on Gaussian mixtures and the MNIST dataset.

**Strengths:**

- **Originality**: This work provides a novel theoretical analysis of how guidance strength affects the classification confidence of VE-SDEs and VP-SDEs, a relationship not previously established in the literature.
- **Quality**: The study is of high quality, featuring a well-organized introduction, a clearly motivated methodology, rigorous theoretical derivations, and well-designed experiments that substantiate the claims.
- **Clarity**: The theoretical findings are well motivated, which makes it easy to follow.
- **Significance**: The research addresses a significant scientific problem by providing a theoretical foundation for the observed performance differences between VP-SDEs and VE-SDEs, which is crucial for their effective application.

**Weaknesses:**

- The claim in Section 5 that VE-SDE outperforms VP-SDE on Gaussian mixtures under strong guidance is not fully substantiated, as the presented analysis itself is independent of the guidance strength.
- While the analysis of diversity is a core contribution, the study lacks empirical experiments that directly evaluate the diversity of generated samples. This omission weakens the central claims regarding the differential behavior of VE-SDE and VP-SDE.
- The difference in classification confidence is presented as the primary reason for VE-SDE's superior performance. However, this may not be the sole factor. The authors should strengthen their argument by demonstrating a strong correlation between their theoretical metric (classification confidence) and established generative quality metrics like FID.
- The numerical experiments, while well-executed on MNIST, would benefit from validation on more complex and diverse image datasets (e.g., CIFAR-10, FFHQ). This would strengthen the generalizability of the conclusions and demonstrate the broader applicability of the theoretical findings.

**Questions:**

- The study would be strengthened by a more comprehensive analysis that directly links the theoretical guidance strength,
$\eta$, to the empirically observed behaviors of mode collapse in both VE-SDE and VP-SDE. A quantitative discussion bridging the derived convergence rates and the diversity of generated samples would be particularly valuable.
- While the theoretical analysis of classification confidence is insightful, its practical importance would be more compelling if supported by empirical evidence. We recommend conducting experiments that demonstrate a clear correlation between the proposed classification confidence metric and established generative performance metrics, such as FID.
- The conclusions are currently based on experiments with the MNIST dataset. To demonstrate the broader applicability and robustness of the findings, it is crucial to include experiments on more challenging and modern image datasets, such as CIFAR-10, CelebA, or FFHQ.
- In Line 198, does $m_{t} = 1$ rather than $m_{t} = 0$ for VE-SDE?

---

> ### Author Response · Authors · 2025-11-19
> **Rebuttal Part 1: Real-world Experiments and Discussion on Diversity**
>
> Thank you for your valuable comments and suggestions. We provide our response to each question below. For the revised paper, we highlight the new part in green.
>
> **Weakness 2 & Weakness 4 & Q3: Larger Real-world experiments and Diversity Quantitative Results (CIFAR10 and CELEBA64).**
>
> Thanks for the valuable comments on real-world experiments. We are more than happy to provide the empirical evidence and quantitative results to support our theoretical guarantee. Despite the MNIST experiments, we conduct experiments on CIFAR-10 and CELEBA64. For CIFAR-10, the target class is the car. For CELEBA64, we divide the dataset into male and female, and the target class is the male image.
>
> - From a qualitative perspective, as shown in **Figure 8 of our revised paper**, VE-based models exhibit better performance and diversity when equipped with a large guidance scale. On the contrary, VP-based models generate samples with low diversity (for example, VP models have a high probability to generate red cars) or distorted images with lower classification confidence (distorted male faces in CelebA64 experiments).
>
> - For the quantitative results of classification, with an MNIST classifier, our paper shows that VE-based models have a higher classification confidence (Figure 7).
>
> - For the diversity, since we focus on the generation with a given label (car for CIFAR 10 and male face for CELCEBA), we can not use the Inception score (which measures the inter-class diversity).
>
>   In this part, we adopt LPIPS diversity [1] to measure the intra-class diversity. More specifically, we generate samples by the model and calculate the LPIPS between samples (higher LPIPS, better diversity). Without any guidance, the LPIPS for VP and VE models are $0.1777$ and $0.1772$, respectively. However, when guidance becomes larger ($\eta=2$), the LPIPS becomes $0.1364$ for VP models, indicating that these models suffer from mode collapse. On the contrary, LPIPS is still $0.1731$ for VE models, which means these models maintain the multi-modal property.
>
> The above qualitative and quantitative results support our theoretical guarantee (faster convergence and better diversity for VE models). We have now added a detailed results and discussion in the revision of our paper (Sec 6, highlighted in green).
>
> **Weakness 1 & Q1: Discussion and Theoretical Guarantee for Diversity.**
>
> Thanks to the kind reviewer for the constructive comments on diversity.  For the title of Section 5, since [2] observes the VP models suffer from mode collapse facing strong guidance, and we observe that VE will not (simulation experiments, our Fig.6), we named this section with the term "facing strong guidance".  As pointed out by the king reviewer, this part discusses the diversity mainly through simulation, real-world experiments, and intuition, and does not involve much analysis on guidance.
>
> For the theoretical guarantee, it is possible to measure the diversity of generated results and compare VP and VE by using the entropy during the denoising process.  In this part, we provide an *intuitive discussion* following the diversity proof of [2]. Let $Q(t, x)$ is the probability density function of $x_t$ (with guidance) $H(t):=-\int Q(t, x) \log Q(t, x) \mathrm{d} x$ be the entropy ( $Q_0(t,x)$ and $H_0(t)$ is the one for $z_t$ without guidance).  As shown in Appendix B.1 of [2], we have that
> $$
> \frac{\partial}{\partial t} H(t)=\sum_{i=1}^d \int\left(1+\frac{\partial^2}{\partial^2 x_i} \log p_{T-t}(x, y)+\eta \frac{\partial^2}{\partial^2 x_i} \log p_{T-t}(y \mid x)\right) Q(t, x) \mathrm{d} x\,,
> $$
> where $\eta$ term is an additional term compared with $\frac{\partial}{\partial t} H_0(t)$. [2] shows that $$\frac{\partial^2}{\partial^2 x_i} \log p_{T-t}(y \mid x)=-\mathrm{tr}[\Sigma\_{T-t}^{-1}\Sigma\_{T-t}^{-1\top}]\mathrm{tr}[M]$$  ($M$ is a matrix corresponds to $\mu_{y'}$ and $q_{T-t}\left(y^{\prime} \mid x\right)$). Since the additional terms is negative, the entropy with guidance will be smaller than without guidance (which leads to a lower diversity). The main difference for VP and VE is $\Sigma_{T-t}$. With $\Sigma =I_d$, $\mathrm{tr}[\Sigma\_{T-t}^{-1}\Sigma\_{T-t}^{-1\top}]=d$ for VP, which is larger that $d/(1+\sigma_{T-t}^2)^2$ for VE in all time. This result indicates that VP with guidance will have a larger negative influence on the entropy and lead to a lower diversity.
>
> We emphasize that the above part is an intuitive discussion instead of a rigorous proof; we leave the rigorous analysis and comparison of VE and VP models with guidance for diversity as a very interesting future work (as shown in our limitations and future work).  To better reflect our observations, we renamed this section as VE Maintain Multi-modal Property During Guidance Process instead of facing strong guidance in our revised paper.

---

> > ### Author Response · Authors · 2025-11-19
> > **Rebuttal Part 2: Question and Reference**
> >
> > **Weakness 3 & Q2: The relationship between classification confidence, diversity, and FID.**
> >
> > As a start, we first recall the definition of FID and classification confidence. For FID, we require the the mean $\mu_g$ and covariance $\Sigma_g$ of generated samples is close to the  real-world data mean $\mu_r$ and $\Sigma_r$ is close enough:
> > $$
> > \mathrm{FID}(p_{\text{real}},p_{\text{gen}})
> > =\|\mu_r-\mu_g\|^2
> > +\mathrm{Tr}\left(\Sigma_r+\Sigma_g-2(\Sigma_r\Sigma_g)^{1/2}\right).
> > $$
> > For the classification confidence, as shown in the following parts, a high classification confidence indicates that $x$ is closer to the mean $\mu_y$. However, this metric does not consider the diversity $\Sigma_g$, which corresponds to the diversity (a small $\Sigma_g$ indicates the sample concentrates around the mean and loses diversity, for example, VP in Fig. 8) and still has a gap with FID
> > $$
> > p(y\mid x)=\frac{w_y \exp(\langle\Sigma^{-1}\mu_y,x\rangle - \frac12 \mu_y^\top\Sigma^{-1}\mu_y)}
> > {\sum_{y'} w_{y'} \exp(\langle\Sigma^{-1}\mu_{y'},x\rangle - \frac12 \mu_{y'}^\top\Sigma^{-1}\mu_{y'})}.
> > $$
> > However, our real-world experiments clearly show that facing a large guidance, VE based models are not only has a higher classification guidance (Fig. 7), but also maintain the diversity (Fig. 8). As a results, VE models will has a lower FID compared to VP when facing large guidance (As shown in the last column Fig.8 of our revised paper,  the distorted images of VP will leads to a high FID).
> >
> >
> >
> > **Q4: A Typo.** Thanks. We have fixed it in our revised paper.
> >
> >
> >
> >
> >
> > [1] Zhu, Jun-Yan, et al. "Toward multimodal image-to-image translation." *Advances in neural information processing systems* 30 (2017).
> >
> > [2] Wu, Yuchen, et al. "Theoretical insights for diffusion guidance: A case study for Gaussian mixture models." *International Conference on Machine Learning*. PMLR, 2024.

---

> > > ### Author Response · Authors · 2025-11-27
> > >
> > > Dear Reviewer c7XG:
> > >
> > > We thank you once again for your careful reading of our paper and your constructive comments and suggestions. In the rebuttal phase, from the empirical perspective, we provide larger real-world experiments (CIFAR10 and CELEBA64) and Diversity Quantitative Results to support our theoretical results.  From the theoretical perspective, following the constructive comments, we provide a preliminary analysis of the entropy.
> > >
> > > We will appreciate it very much if you could let us know whether all your concerns are addressed. We are also more than happy to answer any further questions in the remaining discussion period.
> > >
> > > Best, Authors.

---

### Official Review · Reviewer_a1qN · 2025-11-01

**Soundness:** 2
**Presentation:** 3
**Contribution:** 3
**Rating:** 4
**Confidence:** 3

**Summary:**

The manuscript provides a theoretical analysis on the guidance of Variance Exploding (VE) diffusion model, explaining why VE can achieve state-of-the-art performance in conditional generation tasks. Authors analyze guidance from two perspectives: classification confidence and diversity and prove that though VE’s error is larger than VP without guidance but converges faster with guidance, making it quickly achieve high confidence while maintaining multi-modal structure. Theoretical results are supported the experiments on MNIST, showing VE models achieve higher confidence and accuracy with less distortion under strong guidance.

**Strengths:**

1. Meaningful research questions that address key puzzles within the community;
2. Rigorously logical arguments and solid proofs of theorems.

**Weaknesses:**

1. The experiments seem to be a little weak. Though experiments on MNIST support the derived conclusion, it is too simplistic and lacks diversity. Despite producing the desired results, the evidence remains insufficiently strong. I believe that it would be beneficial if more experiments were included, such as the CIFAR dataset.
2. The assumption of identity variance may be overly restrictive. Even if the marginal distributions are the same, differences across conditional distributions may manifest not only in their means but also in substantial variations in their variances. It would be valuable to examine whether the authors' analysis remains valid under scenarios where the variances of the conditional distributions differ significantly. Similarly, this concern warrants experimental validation. The authors could consider conducting experiments on datasets where conditional distributions exhibit considerable divergence in variance.
3. In Section 5, the authors present a qualitative analysis of the diversity. Though logically coherent, it would be beneficial if authors incorporate quantitative metrics related to diversity—such as the Inception Score—into their experiments. By doing so, the theoretical conclusions derived from the analysis could be further supported from an empirical perspective.
4. Typo: the Gaussian distribution should be expressed as $\mathcal{N}$ in line 260.
5. The assumption seems strong in this paper. The proofs are given under the Mixture of Gaussian distributions rather than a general target distribution. So it is hard to say its practical meaning for complex high-dimensional generation tasks.

**Questions:**

Can the authors give more explanation about the meaning of classification confidence and why they formulate it as Eq. 3?

---

> ### Author Response · Authors · 2025-11-19
> **Rebuttal Part 1: Real-world Experiments and Question**
>
> Thank you for your valuable comments and suggestions. We provide our response to each question below. For the revised paper, we highlight the new part in green.
>
> **Weakness 1 & Weakness 3: Larger Real-world experiments and Diversity Quantitative Results (CIFAR10 and CELEBA64).**
>
> Thanks for the valuable comments on real-world experiments. We are more than happy to provide the empirical evidence and quantitative results to support our theoretical guarantee. Despite the MNIST experiments, we conduct experiments on CIFAR-10 and CELEBA64. For CIFAR-10, the target class is the car. For CELEBA64, we divide the dataset into male and female, and the target class is the male image.
>
> - From a qualitative perspective, as shown in **Figure 8 of our revised paper**, VE-based models exhibit better performance and diversity when equipped with a large guidance scale. On the contrary, VP-based models generate samples with low diversity (for example, VP models have a high probability to generate red cars) or distorted images with lower classification confidence (distorted male faces in CelebA64 experiments).
>
> - For the quantitative results of classification, with an MNIST classifier, our paper shows that VE-based models have a higher classification confidence (Figure 7).
>
> - For the diversity, since we focus on the generation with a given label (car for CIFAR 10 and male face for CELCEBA), we can not use the Inception score (which measures the inter-class diversity).
>
>   In this part, we adopt LPIPS diversity [1] to measure the intra-class diversity. More specifically, we generate samples by the model and calculate the LPIPS between samples (higher LPIPS, better diversity). Without any guidance, the LPIPS for VP and VE models are $0.1777$ and $0.1772$, respectively. However, when guidance becomes larger ($\eta=2$), the LPIPS becomes $0.1364$ for VP models, indicating that these models suffer from mode collapse. On the contrary, LPIPS is still $0.1731$ for VE models, which means these models maintain the multi-modal property.
>
> The above qualitative and quantitative results support our theoretical guarantee (faster convergence and better diversity for VE models). We have now added a detailed results and discussion in the revision of our paper (Sec 6, highlighted in green).
>
> **Weakness 4: A Typo.** Thanks. We have fixed it in our revised paper.
>
> **Q1: The Definition of Classification Confidence.**
>
> Thanks for the question. In this part, we provide a detailed explanation. Eq. (3.1) is the exact closed-form expression of the Bayesian posterior $p(y \mid x)$ under the Gaussian mixture model used in our analysis. For each class $y$, the class-conditional density is
> $$
> p(x\mid y)=\mathcal N(x;\mu_y,\Sigma)
> =\frac{1}{(2\pi)^{d/2}\,|\Sigma|^{1/2}}
> \exp\left(
> -\tfrac12 (x-\mu_y)^\top \Sigma^{-1}(x-\mu_y)
> \right),
> $$
> and the prior is $p(y)=w_y$. Applying Bayes’ rule yields
> $$
> p(y\mid x)
> =\frac{w_y\,p(x\mid y)}{\sum_{y'} w_{y'}\,p(x\mid y')}.
> $$
> By eliminating coefficients and shared $\exp(-1/2 x^{\top}\Sigma^{-1}x)$, we have that:
> $$
> p(y\mid x)=
> \frac{
> w_y \exp(\langle\Sigma^{-1}\mu_y, x\rangle - \tfrac12 \mu_y^\top \Sigma^{-1}\mu_y)
> }{
> \sum_{y'} w_{y'} \exp(\langle\Sigma^{-1}\mu_{y'}, x\rangle - \tfrac12 \mu_{y'}^\top \Sigma^{-1}\mu_{y'})
> }.
> $$
> Thus, Eq. (3.1) is the true posterior probability of label $y$ given the sample $x$ and can be viewed as classification confidence.

---

> ### Author Response · Authors · 2025-11-19
> **Rebuttal Part 2: Extension to General Data**
>
> **Weakness 2&  Weakness 5: Extended Analysis to General Data.**
>
> Thanks for the valuable and helpful comments to broaden the scope of our analysis. We first extend our analysis to a multi-modal GMM with $\Sigma=I$ and conduct simulation experiments on GMMs with different variances to show that VE models still have a higher classification confidence. Then, we discuss a more complex target distribution.
>
> (a) **Extension to multi-modal GMM.** We provide a former corollary (Corollary 5.4 of our revised paper) instead of a remark for the convergence guarantee of multi-modal GMM. More specifically, following exactly the same assumption of [2]:
>
> *Assumption. There exists $\mu_0\in \mathbb{R}^d$ that satisfies (assuming $\mu_y$ is our target modal): (1) for $\forall y'\in \mathcal{Y}$, $\left|\left\langle\mu_y-\mu_0, \mu_{y^{\prime}}-\mu_0\right\rangle\right| \leq \epsilon$ hold for some positive constant $\epsilon$; (2) $ \epsilon \leq\left\\|\mu_y-\mu_0\right\\|_2^2 / 3$; (3) $\Sigma = I_d$.*
>
> We prove that with this assumption, VE models with reverse SDE still have $1-\eta^{-1}(\log \eta)^2$ convergence guarantee, which is still fater than $1-\eta^{-e^{-T}}(\log \eta)^{2 e^{-T}}$ for VP with reverse SDE (under exactly the same assumption). We also note that, though achieving the above extension, the assumption for multi-modal indicates that the mean vectors of each cluster are almost orthogonal to one another and do not influence each other, which simplifies the analysis. It is a very interesting future work to achieve the theoretical guarantee in a more general GMM setting.
>
> *GMM with Different Variance.* In the analysis of 2-GMM and extended multi-modal GMM, we provide the convergence guarantee with $\Sigma=I_d$. However, it is possible for different clusters to have different variances for real-world datasets. By conducting simulation experiments on the $2$-modal GMM with different variance ($\Sigma_1=0.5I_d, \Sigma_2=I_d$, Fig. 5 of our revision paper and $\Sigma_1=2I_d, \Sigma_2=I_d$, Fig. 13), we show that VE-based models still have a faster convergence rate compared with VP models with reverse SDE, which indicates our theoretical guarantee should hold for more general GMM (multi-modal and different variance for each cluster). However, For the different variance, since the (conditional) score has a more complex closed form, we left the theoretical guarantee for GMM with different variance as an interesting and important future work.
> We have now added the above discussion and simulation experiments in the revision of our paper (End of Sec 4.3, highlighted in green).
>
> We have now added the above new results, discussion, and simulation experiments in the revision of our paper (Sec 4.3, highlighted in green).
>
> (b) **Target distribution with low-dimensional modeling.** As the closed-form score of GMM plays an important role in the analysis of convergence guarantee, we aim to maintain the good property and extend it to more general distributions, and we intuitively discuss the next modeling. Since the image data usually admits low-dimensional, a meaningful step is to assume the data has a linear subspace $x=Ax^{LD}$ [3], where $A\in \mathbb{R}^{d\times d^{\mathrm{LD}}}$ (LD means latent dimension). Then, GMM modeling is conducted on the latent subspace $x^{\mathrm{LD}}$. This modeling effectively improves the expressive ability of data distribution, and the score still has a closed-form [3]:
> $$
> \nabla \log p_t(x)= A \nabla \log p_t^{\mathrm{LD}}\left(A^{\top} x\right) -\frac{1}{\sigma_t^2}\left(I_d-A A^{\top}\right)x\,,
> $$
> where $\nabla \log p_t^{\mathrm{LD}}\left(\cdot\right)$ is the latent score function for GMM (whose closed form conditional score and additional guidance have a similar form with Sec. 4.1). Then, the guidance will happen in the latent space. We note that this modeling is closer to the real-world since the latent diffusion models first encode (A nonlinear encoder. However, the analysis with linear subspace is still interesting) the image into the latent subspace, and the diffusion and guidance happen in the latent space. We leave the analysis under such modeling as an interesting future work.
>
>
>
>
>
>
>
> [1] Zhu, Jun-Yan, et al. "Toward multimodal image-to-image translation." *Advances in neural information processing systems* 30 (2017).
>
> [2] Wu, Yuchen, et al. "Theoretical insights for diffusion guidance: A case study for Gaussian mixture models." *International Conference on Machine Learning*. PMLR, 2024.
>
> [3] Chen, Minshuo, et al. "Score approximation, estimation and distribution recovery of diffusion models on low-dimensional data." *International Conference on Machine Learning*. PMLR, 2023.

---

> > ### Author Response · Authors · 2025-11-27
> >
> > Dear Reviewer a1qN:
> >
> > We thank you once again for your careful reading of our paper and your constructive comments and suggestions. In the rebuttal phase, from the empirical perspective, we provide larger real-world experiments (CIFAR10 and CELEBA64) and Diversity Quantitative Results to support our theoretical results. We also provide simulation experiments on different variance and show that VE is still better than VP. From the theoretical perspective, we extend our analysis and results to multi-modal GMM and also discuss how to extend to a more general setting.  We also answer the question about the definition of the classification confidence in the rebuttal phase.
> >
> > We will appreciate it very much if you could let us know whether all your concerns are addressed. We are also more than happy to answer any further questions in the remaining discussion period.
> >
> > Best, Authors.

---

### Official Review · Reviewer_YwNB · 2025-11-01

**Soundness:** 3
**Presentation:** 3
**Contribution:** 3
**Rating:** 4
**Confidence:** 4

**Summary:**

The paper studies guidance for variance-exploding (VE) diffusion models, contrasting them with variance-preserving (VP) models. In a Gaussian-mixture setting, it claims two main results: (i) classification-confidence convergence faster in VE models with guidance strength $\eta$ increasing; (ii) VE’s forward process preserves multi-modality. Simulations and MNIST experiments qualitatively support these findings.

**Strengths:**

The paper studies how guidance influences classification confidence and diversity, which are exactly the two key properties practitioners trade in practice.

Using Gaussian mixture models yields closed-form expressions for scores, enabling detailed proofs and clear diagnostics. The results appear correct and also complements existing study.

The mode-preservation rationale for VE is intuitive and well-motivated.

**Weaknesses:**

The main results focusing on 2-component Gaussian mixture, with a remark at the end promising the direct extension to multiple components. I am ok with Gaussian mixture assumption, however, if the analysis is directly applicable to multiple components, it is recommended to present the multiple-component results.

Many results in the paper is similar to Wu et al., both in setting and presentation. The novelty beyond the VE setup is not very clearly stated.

Experiments are illustrative rather than conclusive. Yet, this is a minor weakness as the majority of the paper is theoretical.

**Questions:**

Theorem 4.2 mainly compares upper bounds and claims that VE is better.

Are there quantitative results in Section 5 to support the intuition? In fact, although VE keeps the mean unchanged, with larger and larger noise added, the three components also become indistinguishable.

Mode collapse is more often used than modal collapse.

I hope the authors can clarify their technical contributions and discuss the results for general finite-component Gaussian mixture models. I am willing to increase my rating if the responses are satisfactory.

---

> ### Author Response · Authors · 2025-11-19
> **Rebuttal Part 1: Extension to General Data**
>
> Thank you for your valuable comments and suggestions. We provide our response to each question below. For the revised paper, we highlight the new part in green.
>
> **Weakness 1& Q1: Extended Analysis to General Data.**
>
> Thanks for the valuable and helpful comments to broaden the scope of our analysis. We first extend our analysis to a multi-modal GMM. Then, we discuss a more complex target distribution.
>
> (a) **Extension to multi-modal GMM.** We provide a former corollary (Corollary 5.4 of our revised paper) instead of a remark for the convergence guarantee of multi-modal GMM. More specifically, following exactly the same assumption of [1]:
>
> *Assumption. There exists $\mu_0\in \mathbb{R}^d$ that satisfies (assuming $\mu_y$ is our target modal): (1) for $\forall y'\in \mathcal{Y}$, $\left|\left\langle\mu_y-\mu_0, \mu_{y^{\prime}}-\mu_0\right\rangle\right| \leq \epsilon$ hold for some positive constant $\epsilon$; (2) $ \epsilon \leq\left\\|\mu_y-\mu_0\right\\|_2^2 / 3$; (3) $\Sigma = I_d$.*
>
> We prove that with this assumption, VE models with reverse SDE still have $1-\eta^{-1}(\log \eta)^2$ convergence guarantee, which is still fater than $1-\eta^{-e^{-T}}(\log \eta)^{2 e^{-T}}$ for VP with reverse SDE (under exactly the same assumption). We also note that, though achieving the above extension, the assumption for multi-modal indicates that the mean vectors of each cluster are almost orthogonal to one another and do not influence each other, which simplifies the analysis. It is a very interesting future work to achieve the theoretical guarantee in a more general GMM setting.
>
> *GMM with Different Variance.* In the analysis of 2-GMM and extended multi-modal GMM, we provide the convergence guarantee with $\Sigma=I_d$. However, it is possible for different clusters to have different variances for real-world datasets. By conducting simulation experiments on the $2$-modal GMM with different variance ($\Sigma_1=0.5I_d, \Sigma_2=I_d$, Fig. 5 of our revision paper and $\Sigma_1=2I_d, \Sigma_2=I_d$, Fig. 13), we show that VE-based models still have a faster convergence rate compared with VP models with reverse SDE, which indicates our theoretical guarantee should hold for more general GMM (multi-modal and different variance for each cluster). However, For the different variance, since the (conditional) score has a more complex closed form, we left the theoretical guarantee for GMM with different variance as an interesting and important future work.
> We have now added the above discussion and simulation experiments in the revision of our paper (End of Sec 4.3, highlighted in green).
>
> We have now added the above new results, discussion, and simulation experiments in the revision of our paper (Sec 4.3, highlighted in green).
>
> (b) **Target distribution with low-dimensional modeling.** As the closed-form score of GMM plays an important role in the analysis of convergence guarantee, we aim to maintain the good property and extend it to more general distributions, and we intuitively discuss the next modeling. Since the image data usually admits low-dimensional, a meaningful step is to assume the data has a linear subspace $x=Ax^{LD}$ [2], where $A\in \mathbb{R}^{d\times d^{\mathrm{LD}}}$ (LD means latent dimension). Then, GMM modeling is conducted on the latent subspace $x^{\mathrm{LD}}$. This modeling effectively improves the expressive ability of data distribution, and the score still has a closed-form [2]:
> $$
> \nabla \log p_t(x)= A \nabla \log p_t^{\mathrm{LD}}\left(A^{\top} x\right) -\frac{1}{\sigma_t^2}\left(I_d-A A^{\top}\right)x\,,
> $$
> where $\nabla \log p_t^{\mathrm{LD}}\left(\cdot\right)$ is the latent score function for GMM (whose closed form conditional score and additional guidance have a similar form with Sec. 4.1). Then, the guidance will happen in the latent space. We note that this modeling is closer to the real-world since the latent diffusion models first encode (A nonlinear encoder. However, the analysis with linear subspace is still interesting) the image into the latent subspace, and the diffusion and guidance happen in the latent space. We leave the analysis under such modeling as an interesting future work.

---

> > ### Author Response · Authors · 2025-11-19
> > **Rebuttal Part 2: Real-world Experiments, Novelty and Contribution**
> >
> > **Weakness 2: Technique Novelty and Challenge.**
> >
> > Thanks to the kind reviewer for the valuable comments to help our work highlight the technique's novelty. Since the GMM setting reflects the multi-modal property and has a closed-form nonlinear score, following [1] (which makes a meaningful step), we adopt this setting to analyze the difference between VP and VE. In the following part, we highlight our technique novelty and contribution.
> >
> > *(a) Analysis for conditional generation without guidance.* Different from [1], we conducted a detailed analysis of the behavior of various models from $\eta=0$ to $\eta=+\infty$ (Sec 4.2 and 4.3 of our work). On the contrary, [1] mainly focuses on convergence guarantee and simply views $\eta=0$ as a baseline (Sec. 4.3 of our work). In Sec. 4.2, we prove the classification confidence without guidance for VP and VE, clearly showing the influence of $T$ and explaining why VE has a poor beginning.
> >
> > *(b) Convergence rate: Make full use of VE property to avoid $\exp{(-T)}$.*
> >
> > For the convergence rate (Sec. 4.3), we make full use of the property of VE models to avoid $\exp{(-T)}$, which is hard for VP due to its forward process design. More specifically, for VP with reverse SDE, we have that (recall that $x_t$ for guidance process, $z_t$ for process without guidance. For simplicity, we reduced the items related to $\eta$ to $\mathcal{G}(\eta)$):
> > $$
> > \mathrm{~d}\left\langle x_t- z_t, \mu_1-\mu_2\right\rangle \geq\left[-\left\langle x_t- z_t, \mu_1-\mu_2\right\rangle+\mathcal{G}_{VP}(\eta)\right] \mathrm{d}t\,,
> > $$
> >
> > where the additional $-\left\langle x_t- z_t, \mu_1-\mu_2\right\rangle$ comes from the calculation $f(T-t)x-g(T-t)^2 s_{T-t}(x,y)$ with constant drifted term $f(t)=1$ and $g^2(T-t)=2$ of VP. To eliminate this term, we need to use the integrating factor method and multiply $\exp{\left(\int 1 \mathrm{d}t\right)}=e^t$ on both sides ($1$ is the coefficient of $-\left\langle x_t- z_t, \mu_1-\mu_2\right\rangle$) and obtain
> > $$
> > \mathrm{~d}\left\langle e^t(x_t- z_t), \mu_1-\mu_2\right\rangle =\left[-\left\langle x_t- z_t, \mu_1-\mu_2\right\rangle+\mathcal{G}_{VP}(\eta)\right] \mathrm{d}t\,,
> > $$
> >
> > where is the source of $e^{-T}$. On the contrary, with $f(t)=0$ and $s_t(x,y)$ for VE, we have that
> > $$
> > \mathrm{~d}\left\langle x_t- z_t, \mu_1-\mu_2\right\rangle =\left[-g(T-t)^2\Sigma_{T-t}^{-1}\left\langle x_t- z_t, \mu_1-\mu_2\right\rangle+\mathcal{G}_{VE}(\eta)\right] \mathrm{d}t\,.
> > $$
> >
> > We know that $\int g(T-t)^2\Sigma_{T-t}^{-1}\mathrm{d}t=\int \frac{1}{\sigma^2+(T-t)}\mathrm{d}t=-\ln{(\sigma^2+(T-t))}$ (using VE (SMLD) as an example) and $\exp{-\ln{(\sigma^2+(T-t))}}=1/(\sigma^2+(T-t))$ will not introduce a additional $\exp{(T)}$ in the final guarantee.
> >
> > *(c) Real World Experiments.* Similar to [1], we first precisely support our theoretical results with simulation experiments. Then, we go beyond the simulation and clearly show the different performance in the real-world experiments.
> >
> > **Weakness 3: Larger Real-world experiments and Quantitative results (CIFAR10 and CELEBA64).**
> >
> > Thanks for the valuable comments on real-world experiments. We are more than happy to provide the empirical evidence and quantitative results to support our theoretical guarantee. Despite the MNIST experiments, we conduct experiments on CIFAR-10 and CELEBA64. For CIFAR-10, the target class is the car. For CELEBA64, we divide the dataset into male and female, and the target class is the male image.
> >
> > - From a qualitative perspective, as shown in **Figure 8 of our revised paper**, VE-based models exhibit better performance and diversity when equipped with a large guidance scale. On the contrary, VP-based models generate samples with low diversity (for example, VP models have a high probability to generate red cars) or distorted images with lower classification confidence (distorted male faces in CelebA64 experiments).
> >
> > - For the quantitative results of classification, with an MNIST classifier, our paper shows that VE-based models have a higher classification confidence (Figure 7).
> >
> > - For the diversity, since we focus on the generation with a given label, we can not use the Inception score (which measures the inter-class diversity). In this part, we adopt LPIPS diversity [3] to measure the intra-class diversity. More specifically, we generate samples and calculate the LPIPS between samples (higher LPIPS, better diversity). Without any guidance, the LPIPS for VP and VE are $0.1777$ and $0.1772$, respectively. However, when guidance becomes larger ($\eta=2$), the LPIPS becomes $0.1364$ for VP, indicating that these models suffer from mode collapse. On the contrary, LPIPS is still $0.1731$ for VE, which means these models maintain the multi-modal property.
> >
> > The above qualitative and quantitative results support our theoretical guarantee (faster convergence and better diversity for VE models). We have now added a detailed results and discussion in the revision of our paper (Sec 6, highlighted in green).

---

> > > ### Author Response · Authors · 2025-11-19
> > > **Rebuttal Part 3: Question and Reference**
> > >
> > > **Q1: Upper Bound Guarantee.**
> > >
> > > This upper bound mainly comes from the bound of $\mathcal{G}(\eta)$ (mentioned in the discussion of the technique). The main order of $T$ (which corresponds to the forward process) has been determined when using the integrating factor method. In this phase, the analysis does not involve the inequality. Our simulation experiments also accurately reflect our theoretical results.
> > >
> > > **Q2: The Quantitative Results in Diversity.**
> > >
> > > The quantitative results in real-world experiments (LPIPS diversity) have shown that VE models have a better ability to avoid mode collapse.
> > >
> > > For the theoretical perspective, though VE and VP forward processes become closer to pure Gaussian as $T$ becomes larger. However, the KL divergence $KL(q_T|\mathcal{N}(0,\sigma_T^2I))$ between the pure Gaussian distribution and forward marginal distribution $p_T$ has a different order for VP and VE models. For VP, the order is $\exp{(-T)}$ (exponential), and the order is $1/\sigma_T^2$ (polynomial) for VE [4]. Hence, facing the same diffusion time $T$, VE has a better property in maintaining multi-mode property compared with VP models.
> > >
> > > **Q3: Mode Collapse.**
> > >
> > > Thanks for the comments. We have used mode collapse instead of modal collapse in our revision paper.
> > >
> > > [1] Wu, Yuchen, et al. "Theoretical insights for diffusion guidance: A case study for Gaussian mixture models." *International Conference on Machine Learning*. PMLR, 2024.
> > >
> > > [2] Chen, Minshuo, et al. "Score approximation, estimation and distribution recovery of diffusion models on low-dimensional data." *International Conference on Machine Learning*. PMLR, 2023.
> > >
> > > [3]Zhu, Jun-Yan, et al. "Toward multimodal image-to-image translation." *Advances in neural information processing systems* 30 (2017).
> > >
> > > [4] Yang, Ruofeng, Bo Jiang, and Shuai Li. "The Polynomial Iteration Complexity for Variance Exploding Diffusion Models: Elucidating SDE and ODE Samplers." *The 28th International Conference on Artificial Intelligence and Statistics*. 2025.

---

> > > > ### Author Response · Authors · 2025-11-27
> > > >
> > > > Dear Reviewer YwNB:
> > > >
> > > > We thank you once again for your careful reading of our paper and your constructive comments and suggestions. In the rebuttal phase, from the empirical perspective, we provide larger real-world experiments (CIFAR10 and CELEBA64) and Diversity Quantitative Results to support our theoretical results. From the theoretical perspective, we extend our analysis and results to multi-modal GMM and also discuss how to extend to a more general setting.  We also discuss our contribution and technique novelty, and answer each question in the rebuttal phase.
> > > >
> > > > We will appreciate it very much if you could let us know whether all your concerns are addressed. We are also more than happy to answer any further questions in the remaining discussion period.
> > > >
> > > > Best, Authors.

---

### Official Review · Reviewer_Yicc · 2025-11-12

**Soundness:** 3
**Presentation:** 3
**Contribution:** 3
**Rating:** 4
**Confidence:** 3

**Summary:**

This paper offers a theoretical and empirical investigation into how guidance operates in variance-exploding (VE) diffusion models. While previous studies have primarily focused on variance-preserving (VP) models, the authors aim to explain why VE-based models, such as EDM, tend to perform better in conditional generation tasks when combined with classifier or classifier-free guidance.

The paper provides analytical results showing that VE diffusion models achieve faster convergence of classification confidence under guidance and preserve greater diversity across samples. The authors derive explicit convergence rates with respect to the guidance strength parameter $\eta$. For the case of VE and VP, the authors proposed and proved their rates, and the result suggests that VE models align more efficiently with the guidance signal.

The authors further argue that VE processes preserve multimodal structure throughout the forward diffusion, which prevents the mode collapse that often occurs in VP models under strong guidance. Experiments on Gaussian mixture models and MNIST support these claims, demonstrating that VE-guided diffusion maintains higher classification confidence and sample diversity compared to VP-guided baselines.

**Strengths:**

The motivation is clear and relevant. Understanding why VE models perform better under guidance is important, as these models dominate practical diffusion-based generative systems such as EDM and SMLD. The paper fills a gap in theoretical understanding. The mathematical analysis is rigorous and well structured. The derivations of convergence rates are transparent and carefully derived under clearly stated assumptions. The extension of the analysis from VP to VE models is technically sound. The study deepens understanding of guidance in diffusion models and could influence future work on improving conditional generation or training strategies in VE-based systems.

**Weaknesses:**

The paper’s theoretical results are derived under restrictive assumptions, primarily Gaussian mixture models. While this setting makes the analysis tractable, it limits the generality of the conclusions. Real-world data are rarely Gaussian, and it remains uncertain how the theoretical findings extend to complex, high-dimensional distributions used in practical diffusion models.

The discussion of diversity preservation is mostly qualitative and lacks a formal definition or measurable bound. A quantitative diversity metric or theoretical result would make the argument more convincing. The empirical evaluation, though consistent with the theory, is limited to small datasets such as MNIST and synthetic Gaussian mixtures. The experiments do not show whether the same patterns appear in large-scale or multimodal diffusion models used in realistic image or text-to-image generation tasks.

Finally, the contribution is solid but still kind of incremental in scope. The paper primarily adapts established VP analyses to the VE case rather than introducing an entirely new theoretical framework.

**Questions:**

1. The theoretical analysis assumes Gaussian mixtures. How do you expect the conclusions to generalize to non-Gaussian or multimodal real-world data distributions, such as those seen in text-to-image models?
2. Does the faster convergence rate hold for both classifier and classifier-free guidance, or is it specific to one type?
3. Could larger-scale experiments on more complex datasets confirm that VE-guided diffusion retains higher diversity and faster convergence beyond the toy examples?

---

> ### Author Response · Authors · 2025-11-19
> **Rebuttal Part 1: Extension to General Data**
>
> Thank you for your valuable comments and suggestions. We provide our response to each question below. For the revised paper, we highlight the new part in green.
>
> **Weakness 1& Q1: Extended Analysis to General Data.**
>
> Thanks for the valuable and helpful comments to broaden the scope of our analysis. We first extend our analysis to a multi-modal GMM. Then, we discuss a more complex target distribution.
>
> (a) **Extension to multi-modal GMM.** We provide a former corollary (Corollary 5.4 of our revised paper) instead of a remark for the convergence guarantee of multi-modal GMM. More specifically, following exactly the same assumption of [1]:
>
> *Assumption. There exists $\mu_0\in \mathbb{R}^d$ that satisfies (assuming $\mu_y$ is our target modal): (1) for $\forall y'\in \mathcal{Y}$, $\left|\left\langle\mu_y-\mu_0, \mu_{y^{\prime}}-\mu_0\right\rangle\right| \leq \epsilon$ hold for some positive constant $\epsilon$; (2) $ \epsilon \leq\left\\|\mu_y-\mu_0\right\\|_2^2 / 3$; (3) $\Sigma = I_d$.*
>
> We prove that with this assumption, VE models with reverse SDE still have $1-\eta^{-1}(\log \eta)^2$ convergence guarantee, which is still fater than $1-\eta^{-e^{-T}}(\log \eta)^{2 e^{-T}}$ for VP with reverse SDE (under exactly the same assumption). We also note that, though achieving the above extension, the assumption for multi-modal indicates that the mean vectors of each cluster are almost orthogonal to one another and do not influence each other, which simplifies the analysis. It is a very interesting future work to achieve the theoretical guarantee in a more general GMM setting.
>
> *GMM with Different Variance.* In the analysis of 2-GMM and extended multi-modal GMM, we provide the convergence guarantee with $\Sigma=I_d$. However, it is possible for different clusters to have different variances for real-world datasets. By conducting simulation experiments on the $2$-modal GMM with different variance ($\Sigma_1=0.5I_d, \Sigma_2=I_d$, Fig. 5 of our revision paper and $\Sigma_1=2I_d, \Sigma_2=I_d$, Fig. 13), we show that VE-based models still have a faster convergence rate compared with VP models with reverse SDE, which indicates our theoretical guarantee should hold for more general GMM (multi-modal and different variance for each cluster). However, For the different variance, since the (conditional) score has a more complex closed form, we left the theoretical guarantee for GMM with different variance as an interesting and important future work.
> We have now added the above discussion and simulation experiments in the revision of our paper (End of Sec 4.3, highlighted in green).
>
> We have now added the above new results, discussion, and simulation experiments in the revision of our paper (Sec 4.3, highlighted in green).
>
> (b) **Target distribution with low-dimensional modeling.** As the closed-form score of GMM plays an important role in the analysis of convergence guarantee, we aim to maintain the good property and extend it to more general distributions, and we intuitively discuss the next modeling. Since the image data usually admits low-dimensional, a meaningful step is to assume the data has a linear subspace $x=Ax^{LD}$ [2], where $A\in \mathbb{R}^{d\times d^{\mathrm{LD}}}$ (LD means latent dimension). Then, GMM modeling is conducted on the latent subspace $x^{\mathrm{LD}}$. This modeling effectively improves the expressive ability of data distribution, and the score still has a closed-form [2]:
> $$
> \nabla \log p_t(x)= A \nabla \log p_t^{\mathrm{LD}}\left(A^{\top} x\right) -\frac{1}{\sigma_t^2}\left(I_d-A A^{\top}\right)x\,,
> $$
> where $\nabla \log p_t^{\mathrm{LD}}\left(\cdot\right)$ is the latent score function for GMM (whose closed form conditional score and additional guidance have a similar form with Sec. 4.1). Then, the guidance will happen in the latent space. We note that this modeling is closer to the real-world since the latent diffusion models first encode (A nonlinear encoder. However, the analysis with linear subspace is still interesting) the image into the latent subspace, and the diffusion and guidance happen in the latent space. We leave the analysis under such modeling as an interesting future work.

---

> > ### Author Response · Authors · 2025-11-19
> > **Rebuttal Part 2: Real-world Experiments and Novelty**
> >
> > **Weakness 2& Q3: Larger Real-world experiments and Diversity Quantitative Results (CIFAR10 and CELEBA64).**
> >
> > Thanks for the valuable comments on real-world experiments. We are more than happy to provide the empirical evidence and quantitative results to support our theoretical guarantee. Despite the MNIST experiments, we conduct experiments on CIFAR-10 and CELEBA64. For CIFAR-10, the target class is the car. For CELEBA64, we divide the dataset into male and female, and the target class is the male image.
> >
> > - From a qualitative perspective, as shown in **Figure 8 of our revised paper**, VE-based models exhibit better performance and diversity when equipped with a large guidance scale. On the contrary, VP-based models generate samples with low diversity (for example, VP models have a high probability to generate red cars) or distorted images with lower classification confidence (distorted male faces in CelebA64 experiments).
> >
> > - For the quantitative results of classification, with an MNIST classifier, our paper shows that VE-based models have a higher classification confidence (Figure 7).
> >
> > - For the diversity, since we focus on the generation with a given label, we can not use the Inception score (which measures the inter-class diversity). In this part, we adopt LPIPS diversity [3] to measure the intra-class diversity. More specifically, we generate samples and calculate the LPIPS between samples (higher LPIPS, better diversity). Without any guidance, the LPIPS for VP and VE are $0.1777$ and $0.1772$, respectively. However, when guidance becomes larger ($\eta=2$), the LPIPS becomes $0.1364$ for VP, indicating that these models suffer from mode collapse. On the contrary, LPIPS is still $0.1731$ for VE, which means these models maintain the multi-modal property.
> >
> > The above results support our theoretical guarantee (faster convergence and better diversity for VE models). We have now added a detailed results and discussion in the revision of our paper (Sec 6, highlighted in green).
> >
> > **Weakness 3: Technique Novelty and Challenge.**
> >
> > Thanks to the kind reviewer for the helpful comments to help our work highlight the technique's novelty. Since the GMM setting reflects the multi-modal property and has a closed-form nonlinear score, following [1] (which makes a meaningful step), we adopt this setting to analyze the difference between VP and VE. In the following part, we highlight our technique novelty and contribution.
> >
> > *(a) Analysis for conditional generation without guidance.* Different from [1], we conducted a detailed analysis of the behavior of various models from $\eta=0$ to $\eta=+\infty$ (Sec 4.2 and 4.3 of our work). On the contrary, [1] mainly focuses on convergence guarantee and simply views $\eta=0$ as a baseline (Sec. 4.3 of our work). In Sec. 4.2, we prove the classification confidence without guidance for VP and VE models, clearly showing the influence of $T$ and explaining why VE has a poor beginning.
> >
> > *(b) Convergence rate: Make full use of VE property to avoid $\exp{(-T)}$.*
> >
> > For the convergence rate (Sec. 4.3), we make full use of the property of VE models to avoid $\exp{(-T)}$, which is hard for VP due to its forward process design. More specifically, for VP based models with reverse SDE, we have that (recall that $x_t$ for guidance process, $z_t$ for process without guidance. For simplicity, we reduced the items related to $\eta$ to $\mathcal{G}(\eta)$):
> > $$
> > \mathrm{~d}\left\langle x_t- z_t, \mu_1-\mu_2\right\rangle \geq\left[-\left\langle x_t- z_t, \mu_1-\mu_2\right\rangle+\mathcal{G}_{VP}(\eta)\right] \mathrm{d}t\,,
> > $$
> >
> > where the additional $-\left\langle x_t- z_t, \mu_1-\mu_2\right\rangle$ comes from the calculation $f(T-t)x-g(T-t)^2 s_{T-t}(x,y)$ with constant drifted term $f(t)=1$ and $g^2(T-t)=2$ of VP. To eliminate this term, we need to use the integrating factor method and multiply $\exp{\left(\int 1 \mathrm{d}t\right)}=e^t$ on both sides ($1$ is the coefficient of $-\left\langle x_t- z_t, \mu_1-\mu_2\right\rangle$) and obtain
> > $$
> > \mathrm{~d}\left\langle e^t(x_t- z_t), \mu_1-\mu_2\right\rangle =\left[-\left\langle x_t- z_t, \mu_1-\mu_2\right\rangle+\mathcal{G}_{VP}(\eta)\right] \mathrm{d}t\,,
> > $$
> >
> > where is the source of $e^{-T}$. On the contrary, with $f(t)=0$ and $s_t(x,y)$ for VE, we have that
> > $$
> > \mathrm{~d}\left\langle x_t- z_t, \mu_1-\mu_2\right\rangle =\left[-g(T-t)^2\Sigma_{T-t}^{-1}\left\langle x_t- z_t, \mu_1-\mu_2\right\rangle+\mathcal{G}_{VE}(\eta)\right] \mathrm{d}t\,.
> > $$
> >
> > We know that $\int g(T-t)^2\Sigma_{T-t}^{-1}\mathrm{d}t=\int \frac{1}{\sigma^2+(T-t)}\mathrm{d}t=-\ln{(\sigma^2+(T-t))}$ (using VE (SMLD) as an example) and $\exp{(-\ln{(\sigma^2+(T-t))})}=1/(\sigma^2+(T-t))$ will not introduce a additional $\exp{(T)}$ in the final guarantee.
> >
> > *(c) Real World Experiments.* Similar to [1], we first precisely support our theoretical results with simulation experiments. Then, we go beyond the simulation and clearly show the different performance in the real-world experiments.

---

> > > ### Author Response · Authors · 2025-11-19
> > > **Rebuttal Part 3: Question and Reference**
> > >
> > > **Q2: Theory for classifier and classifier-free guidance.**
> > >
> > > As shown at the end of Sec. 3, with the closed-form score of GMM, the $x_t$ of classifier guidance and cfg is exactly the same, which indicates that our analysis holds for the two methods.
> > >
> > > [1] Wu, Yuchen, et al. "Theoretical insights for diffusion guidance: A case study for Gaussian mixture models." *International Conference on Machine Learning*. PMLR, 2024.
> > >
> > > [2] Chen, Minshuo, et al. "Score approximation, estimation and distribution recovery of diffusion models on low-dimensional data." *International Conference on Machine Learning*. PMLR, 2023.
> > >
> > > [3] Zhu, Jun-Yan, et al. "Toward multimodal image-to-image translation." *Advances in neural information processing systems* 30 (2017).

---

> > > > ### Author Response · Authors · 2025-11-27
> > > >
> > > > Dear Reviewer Yicc:
> > > >
> > > > We thank you once again for your careful reading of our paper and your constructive comments and suggestions. In the rebuttal phase, from the empirical perspective, we provide larger real-world experiments (CIFAR10 and CELEBA64) and Diversity Quantitative Results to support our theoretical results. From the theoretical perspective, we extend our analysis and results to multi-modal GMM and also discuss how to extend to a more general setting. We also disucss our contribution and technique novelty.
> > > >
> > > > We will appreciate it very much if you could let us know whether all your concerns are addressed. We are also more than happy to answer any further questions in the remaining discussion period.
> > > >
> > > > Best, Authors.

---

### Author Response · Authors · 2025-12-03
**Summary of Rebuttal**

Dear Area Chair,

Thank you very much for taking over the assessment of our submission. To facilitate a quick understanding of our work and rebuttal, we provide a concise summary below.

First of all, we thank the kind reviewers for their constructive comments. We thank reviewers for the recognition of Soundness, Presentation, and Contribution (both rating 3 Good from Reviewer Yicc, YwNB, and c7XG, except a rating 2 of Soundness from Reviewer a1qN) of our work. We are also happy to summarize the shared constructive comments of reviewers and briefly show how we addressed these concerns in the revised paper  (highlighted in green) and rebuttal.

The shared concerns are that (a) conducting larger-scale experiments   (for example, CIFAR10 and CLELEBA64)  and providing quantitative results support our theoretical results; (b) extending our analysis on $2$-mode GMM to the multi-modal GMM (even more general data distribution). We have provided additional experiments (sec. 6) and theory (sec. 4.3) in our revised paper (highlighted in green) and in the rebuttal comments. We also provide responses for each question and the weaknesses of reviewers in the rebuttal phase.

In the following part, we briefly summarize the feedback for the common concerns and our contribution.

**(a) Larger-scale real-world experiments and Quantitative Results**

We provide the empirical evidence and quantitative results to support our theoretical guarantee. Despite the MNIST experiments, we conduct experiments on CIFAR-10 and CELEBA64. For CIFAR-10, the target class is the car. For CELEBA64, the target class is the male image.

- From a qualitative perspective, as shown in **Figure 8 of our revised paper**, VE-based models exhibit better performance and diversity when equipped with a large guidance scale. On the contrary, VP-based models generate samples with low diversity or distorted images with lower classification confidence.

- For the quantitative results, with an MNIST classifier, our paper shows that VE-based models have a higher classification confidence on MNIST (Figure 7). For the diversity, we adopt LPIPS diversity to measure the intra-class diversity (higher is better). Without any guidance, the LPIPS for VP and VE models is close. However, when guidance becomes larger, the LPIPS for VP becomes much smaller compared to VE, indicating that VP suffers from mode collapse.

**(b) Extended Theoretical Results.**

In the rebuttal phase, we first extend our analysis to a multi-modal GMM. Then, we discuss a more complex target distribution.

*Extension to multi-modal GMM.* We provide a formal corollary (Corollary 5.4 of our revised paper) for the convergence guarantee of multi-modal GMM (with the same $I_d$ variance for each modal). We prove that with this assumption, VE models with reverse SDE still have $1-\eta^{-1}(\log \eta)^2$ convergence guarantee, which is still fater than $1-\eta^{-e^{-T}}(\log \eta)^{2 e^{-T}}$ for VP with reverse SDE (under exactly the same assumption). After that, we also discuss the influence of different variances of each modal through simulation experiments and show that under reverse SDE, VE-based models still have a faster convergence rate compared with VP models, which further supports our theoretical guarantee.

*Target distribution with low-dimensional modeling.* After that, we also further discuss how to extend the analysis to more general distributions based on the low-dimensional property of real-world data (details in the rebuttal part).

**(c) Contribution and Technique Contribution.**

This work takes the first step in analyzing the difference between VP and VE models when facing different scale guidance, which is helpful in deepening the understanding of conditional generation tasks with different models. In each step of the analysis, we have (simulation and real-world) experiments to support our theoretical guarantee and clearly show why VE has a poor beginning but has a faster convergence.

*(a) Analysis for conditional generation without guidance.* Different from previous work, we conducted a detailed analysis of the behavior of various models from $\eta=0$ to $\eta=+\infty$ (Sec 4.2 and 4.3 of our work). On the contrary, previous work mainly focuses on convergence guarantee and simply views $\eta=0$ as a baseline (Sec. 4.3 of our work). In Sec. 4.2, we prove the classification confidence without guidance for VP and VE models, clearly showing the influence of $T$ and explaining why VE has a poor beginning.

*(b) Convergence rate.*

For the convergence rate (Sec. 4.3), we make full use of the property of VE models to avoid $\exp{(-T)}$, which is hard for VP due to its forward process design (details in the rebuttal discussion).

*(c) Real World Experiments.* Similar to previous work, we first precisely support our theoretical results with simulation experiments. Then, we go beyond the simulation and clearly show the different performance in the real-world experiments.

---

### Meta-Review · Area_Chair_KznE · 2026-01-12

**Summary:**

The paper studies guidance for variance-exploding diffusion models, compare with variance-preserving models. The reviewers acknowledged the paper's clear motivation. However, the consensus was that the work, in its current form, is marginally below the acceptance threshold for ICLR. The primary concerns are:

The core theoretical results are derived under restrictive assumptions. Three Reviewers expressed doubt about how these findings extend to complex, high-dimensional, real-world data distributions where these assumptions rarely hold.

Reviewer YwNB noted the paper's close similarity in setting and analysis to prior work (Wu et al., 2024), and the novelty of adapting the analysis from VP to VE was not sufficiently compelling on its own.(Reviewer Yicc).

Reviewers requested larger-scale experiments on more modern datasets.

**Reviewer Concerns:**

Addressed Concerns:

The authors provided new experiments on CIFAR-10 and CelebA-64, along with a quantitative diversity metric.

Outstanding Concerns:

The analysis remains restricted to simplified GMM assumptions. Proposed extensions to general distributions are speculative, leaving the practical relevance unconvincing.

The work is still viewed as a direct adaptation of an existing VP analysis to VE, with insufficient novelty for the conference bar.

**Reviewer Scores:**

Reviewer Yicc (Initial: 4): Would likely maintain a score of 4.

Reviewer YwNB (Initial: 4): Would likely maintain a score of 4.

Reviewer a1qN (Initial: 4): Would likely maintain a score of 4.

Reviewer c7XG (Initial: 6): Would likely decrease to a 5.

---

### Decision · Program_Chairs · 2026-01-26

Reject